# MMA Training: Direct Input Space Margin Maximization through Adversarial Training

**Gavin Weiguang Ding**[1], **Yash Sharma**[2,3] *, **Kry Yik Chau Lui**[1], **Ruitong Huang**[1]
[1]Borealis AI  [2]University of Tuebingen  [3]Max Planck Institute for Intelligent Systems

## Abstract

We study adversarial robustness of neural networks from a margin maximization perspective, where margins are defined as the distances from inputs to a classifier's decision boundary. Our study shows that maximizing margins can be achieved by minimizing the adversarial loss on the decision boundary at the "shortest successful perturbation", demonstrating a close connection between adversarial losses and the margins. We propose Max-Margin Adversarial (MMA) training to directly maximize the margins to achieve adversarial robustness. Instead of adversarial training with a fixed $\epsilon$, MMA offers an improvement by enabling adaptive selection of the "correct" $\epsilon$ as the margin individually for each data point. In addition, we rigorously analyze adversarial training with the perspective of margin maximization, and provide an alternative interpretation for adversarial training, maximizing either a lower or an upper bound of the margins. Our experiments empirically confirm our theory and demonstrate MMA training's efficacy on the MNIST and CIFAR10 datasets w.r.t. $\ell_\infty$ and $\ell_2$ robustness. Code and models are available at https://github.com/BorealisAI/mma_training.

## 1 Introduction

Despite their impressive performance on various learning tasks, neural networks have been shown to be vulnerable to adversarial perturbations (Szegedy et al., 2013; Biggio et al., 2013). An artificially constructed imperceptible perturbation can cause a significant drop in the prediction accuracy of an otherwise accurate network. The level of distortion is measured by the magnitude of the perturbations (e.g. in $\ell_\infty$ or $\ell_2$ norms), i.e. the distance from the original input to the perturbed input. Figure 1 shows an example, where the classifier changes its prediction from panda to bucket when the input is perturbed from the blue sample point to the red one. Figure 1 also shows the natural connection between adversarial robustness and the

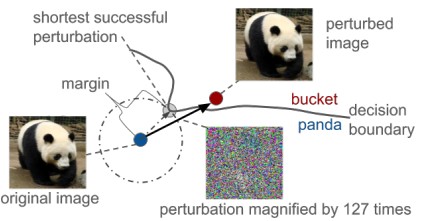

Figure 1: Illustration of decision boundary, margin, and shortest successful perturbation on application of an adversarial perturbation.

margins of the data points, where the margin is defined as the distance from a data point to the classifier's decision boundary. Intuitively, the margin of a data point is the minimum distance that $x$ has to be perturbed to change the classifier's prediction. Thus, the larger the margin is, the farther the distance from the input to the decision boundary is, the more robust the classifier is w.r.t. this input.

Although naturally connected to adversarial robustness, "directly" maximizing margins has not yet been thoroughly studied in the adversarial robustness literature. Instead, the method of minimax adversarial training (Huang et al., 2015; Madry et al., 2017) is arguably the most common defense to adversarial perturbations due to its effectiveness and simplicity. Adversarial training attempts to minimize the maximum loss within a fixed sized neighborhood about the training data using projected gradient descent (PGD). Despite advancements made in recent years (Hendrycks et al., 2019; Zhang et al., 2019a; Shafahi et al., 2019; Zhang et al., 2019b; Stanforth et al., 2019; Carmon et al., 2019), adversarial training still suffers from a fundamental problem, the perturbation length $\epsilon$ has to be set and is fixed throughout the training process. In general, the setting of $\epsilon$ is arbitrary, based on assumptions on whether perturbations within the defined ball are "imperceptible" or not. Recent work (Guo et al., 2018; Sharma et al., 2019) has demonstrated that these assumptions do not

---

*Work done at Borealis AI.

consistently hold true, commonly used $\epsilon$ settings assumed to only allow imperceptible perturbations in fact do not. If $\epsilon$ is set too small, the resulting models lack robustness, if too large, the resulting models lack in accuracy. Moreover, individual data points may have different intrinsic robustness, the variation in ambiguity in collected data is highly diverse, and fixing one $\epsilon$ for all data points across the whole training procedure is likely suboptimal.

Instead of improving adversarial training with a fixed perturbation magnitude, we revisit adversarial robustness from the margin perspective, and propose Max-Margin Adversarial (MMA) training for "direct" input margin maximization. By directly maximizing margins calculated for each data point, MMA training allows for optimizing the "***current*** robustness" of the data, the "***correct***" $\epsilon$ at this point in training for each sample ***individually***, instead of robustness w.r.t. a predefined magnitude.

While it is intuitive that one can achieve the greatest possible robustness by maximizing the margin of a classifier, this maximization has technical difficulties. In Section 2, we overcome these difficulties and show that margin maximization can be achieved by minimizing a classification loss w.r.t. model parameters, at the "shortest successful perturbation". This makes gradient descent viable for margin maximization, despite the fact that model parameters are entangled in the constraints.

We further analyze adversarial training (Huang et al., 2015; Madry et al., 2017) from the perspective of margin maximization in Section 3. We show that, for each training example, adversarial training with fixed perturbation length $\epsilon$ is maximizing a lower (or upper) bound of the margin, if $\epsilon$ is smaller (or larger) than the margin of that training point. As such, MMA training improves adversarial training, in the sense that it selects the "correct" $\epsilon$, the margin value for each example.

Finally in Section 4, we test and compare MMA training with adversarial training on MNIST and CIFAR10 w.r.t. $\ell_\infty$ and $\ell_2$ robustness. Our method achieves higher robustness accuracies on average under a variety of perturbation magnitudes, which echoes its goal of maximizing the average margin. Moreover, MMA training automatically balances accuracy vs robustness while being insensitive to its hyperparameter setting, which contrasts sharply with the sensitivity of standard adversarial training to its fixed perturbation magnitude. MMA trained models not only match the performance of the best adversarially trained models with carefully chosen training $\epsilon$ under different scenarios, it also matches the performance of ensembles of adversarially trained models.

In this paper, we focus our theoretical efforts on the formulation for directly maximizing the input space margin, and understanding the standard adversarial training method from a margin maximization perspective. We focus our empirical efforts on thoroughly examining our MMA training algorithm, comparing with adversarial training with a fixed perturbation magnitude.

## 1.1 RELATED WORKS

Although not often explicitly stated, many defense methods are related to increasing the margin. One class uses regularization to constrain the model's Lipschitz constant (Cisse et al., 2017; Ross & Doshi-Velez, 2017; Hein & Andriushchenko, 2017; Sokolic et al., 2017; Tsuzuku et al., 2018), thus samples with small loss would have large margin since the loss cannot increase too fast. If the Lipschitz constant is merely regularized at the data points, it is often too local and not accurate in a neighborhood. When globally enforced, the Lipschitz constraint on the model is often too strong that it harms accuracy. So far, such methods have not achieved strong robustness. There are also efforts using first-order approximation to estimate and maximize input space margin (Matyasko & Chau, 2017; Elsayed et al., 2018; Yan et al., 2019). Similar to local Lipschitz regularization, the reliance on local information often does not provide accurate margin estimation and efficient maximization. Such approaches have also not achieved strong robustness so far. Croce et al. (2018) aim to enlarge the linear region around a input example, such that the nearest point, to the input and on the decision boundary, is inside the linear region. Here, the margin can be calculated analytically and hence maximized. However, the analysis only works on ReLU networks, and the implementation so far only works on improving robustness under small perturbations. We defer some detailed discussions on related works to Appendix B, including a comparison between MMA training and SVM.

## 1.2 NOTATIONS AND DEFINITIONS

We focus on $K$-class classification problems. Denote $\mathcal{S} = \{x_i, y_i\}$ as the training set of input-label data pairs sampled from data distribution $\mathcal{D}$. We consider the classifier as a score function

$\boldsymbol{f}_\theta(x) = \left(f_\theta^1(x), \ldots, f_\theta^K(x)\right)$, parametrized by $\theta$, which assigns score $f_\theta^i(x)$ to the $i$-th class. The predicted label of $x$ is then decided by $\hat{y} = \arg\max_i f_\theta^i(x)$.

Let $L_\theta^{01}(x, y) = \mathbb{I}(\hat{y} \neq y)$ be the 0-1 loss indicating classification error, where $\mathbb{I}(\cdot)$ is the indicator function. For an input $(x, y)$, we define its margin w.r.t. the classifier $\boldsymbol{f}_\theta(\cdot)$ as:

$$d_\theta(x, y) = \|\delta^*\| = \min \|\delta\| \quad \text{s.t.} \ \delta : L_\theta^{01}(x + \delta, y) = 1, \tag{1}$$

where $\delta^* = \arg\min_{L_\theta^{01}(x+\delta,y)=1} \|\delta\|$ is the "shortest successful perturbation". We give an equivalent definition of margin with the *"logit margin loss"* $L_\theta^{\text{LM}}(x, y) = \max_{j \neq y} f_\theta^j(x) - f_\theta^y(x)$. [1] The level set $\{x : L_\theta^{\text{LM}}(x, y) = 0\}$ corresponds to the decision boundary of class $y$. Also, when $L_\theta^{\text{LM}}(x, y) < 0$, the classification is correct, and when $L_\theta^{\text{LM}}(x, y) \geq 0$, the classification is wrong. Therefore, we can define the margin in Eq. (1) in an equivalent way by $L_\theta^{\text{LM}}(\cdot)$ as:

$$d_\theta(x, y) = \|\delta^*\| = \min \|\delta\| \quad \text{s.t.} \ \delta : L_\theta^{\text{LM}}(x + \delta, y) \geq 0, \tag{2}$$

where $\delta^* = \arg\min_{L_\theta^{\text{LM}}(x+\delta,y)\geq 0} \|\delta\|$ is again the "shortest successful perturbation". For the rest of the paper, we use the term "margin" to denote $d_\theta(x, y)$ in Eq. (2). For other notions of margin, we will use specific phrases, e.g. "SLM-margin" or "logit margin."

## 2 MAX-MARGIN ADVERSARIAL TRAINING

We propose to improve adversarial robustness by maximizing the average margin of the data distribution $\mathcal{D}$, called Max-Margin Adversarial (MMA) training, by optimizing the following objective:

$$\min_\theta \{ \sum_{i \in \mathcal{S}_\theta^+} \max\{0, d_{\max} - d_\theta(x_i, y_i)\} + \beta \sum_{j \in \mathcal{S}_\theta^-} \mathcal{J}_\theta(x_j, y_j) \}, \tag{3}$$

where $\mathcal{S}_\theta^+ = \{i : L_\theta^{\text{LM}}(x_i, y_i) < 0\}$ is the set of correctly classified examples, $\mathcal{S}_\theta^- = \{i : L_\theta^{\text{LM}}(x_i, y_i) \geq 0\}$ is the set of wrongly classified examples, $\mathcal{J}_\theta(\cdot)$ is a regular classification loss function, e.g. cross entropy loss, $d_\theta(x_i, y_i)$ is the margin for correctly classified samples, and $\beta$ is the coefficient for balancing correct classification and margin maximization. Note that the margin $d_\theta(x_i, y_i)$ is inside the hinge loss with threshold $d_{\max}$ (a hyperparameter), which forces the learning to focus on the margins that are smaller than $d_{\max}$. Intuitively, MMA training simultaneously minimizes classification loss on wrongly classified points in $\mathcal{S}_\theta^-$ and maximizes the margins of correctly classified points in $d_\theta(x_i, y_i)$ until it reaches $d_{\max}$. Note that we do not maximize margins on wrongly classified examples. Minimizing the objective in Eq. (3) turns out to be a technical challenge. While $\nabla_\theta \mathcal{J}_\theta(x_j, y_j)$ can be easily computed by standard back-propagation, computing the gradient of $d_\theta(x_i, y_i)$ needs some technical developments.

In the next section, we show that margin maximization can still be achieved by minimizing a classification loss w.r.t. model parameters, at the "shortest successful perturbation". For smooth functions, a stronger result exists: the gradient of the margin w.r.t. model parameters can be analytically calculated, as a scaled gradient of the loss. Such results make gradient descent viable for margin maximization, despite the fact that model parameters are entangled in the constraints.

### 2.1 MARGIN MAXIMIZATION

Recall that
$$d_\theta(x, y) = \|\delta^*\| = \min \|\delta\| \quad \text{s.t.} \ \delta : L_\theta^{\text{LM}}(x + \delta, y) \geq 0.$$
Note that the constraint of the above optimization problem depends on model parameters, thus margin maximization is a max-min nested optimization problem with a parameter-dependent constraint in its inner minimization. [2] Computing such gradients for a linear model is easy due to the existence of its closed-form solution, e.g. SVM, but it is not so for general functions such as neural networks.

The next theorem provides a viable way to increase $d_\theta(x, y)$.

**Theorem 2.1.** *Gradient descent on $L_\theta^{LM}(x + \delta^*, y)$ w.r.t. $\theta$ with a proper step size increases $d_\theta(x, y)$, where $\delta^* = \arg\min_{L_\theta^{LM}(x+\delta,y)\geq 0} \|\delta\|$ is the shortest successful perturbation given the current $\theta$.*

---

[1] Since the scores $\left(f_\theta^1(x), \ldots, f_\theta^K(x)\right)$ output by $\boldsymbol{f}_\theta$ are also called logits in neural network literature.

[2] In adversarial training (Madry et al., 2017), the constraint on the inner max does NOT have such problem.

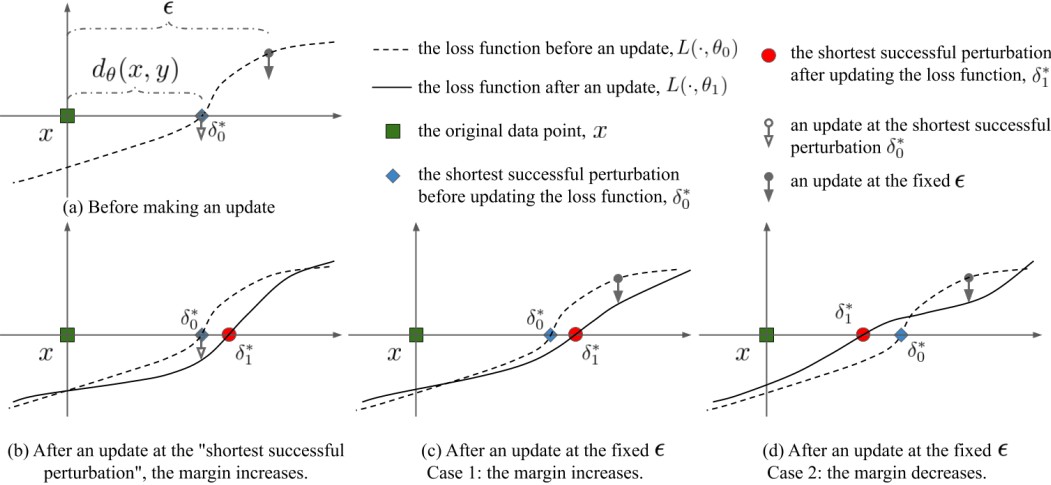

Figure 2: A 1-D example on how margin is affected by decreasing the loss at different locations.

Theorem 2.1 summarizes the theoretical results, where we show separately later

    1) how to calculate the gradient of the margin under some smoothness assumptions;

    2) without smoothness, margin maximization can still be achieved by minimizing the loss at the shortest successful perturbation.

**Calculating gradients of margins for smooth loss and norm:** Denote $L_\theta^{\mathrm{LM}}(x + \delta, y)$ by $L(\delta, \theta)$ for brevity. It is easy to see that for a wrongly classified example $(x, y)$, $\delta^*$ is achieved at $\mathbf{0}$ and thus $\nabla_\theta d_\theta(x, y) = 0$. Therefore we focus on correctly classified examples. Denote the Lagrangian as $\mathcal{L}_\theta(\delta, \lambda) = \|\delta\| + \lambda L(\delta, \theta)$. For a fixed $\theta$, denote the optimizers of $\mathcal{L}_\theta(\delta, \lambda)$ by $\delta^*$ and $\lambda^*$. The following theorem shows how to compute $\nabla_\theta d_\theta(x, y)$.

**Proposition 2.1.** *Let $\epsilon(\delta) = \|\delta\|$. Given a fixed $\theta$, assume that $\delta^*$ is unique, $\epsilon(\delta)$ and $L(\delta, \theta)$ are $C^2$ functions in a neighborhood of $(\delta^*, \theta)$, and the matrix* $\begin{pmatrix} \frac{\partial^2 \epsilon(\delta^*)}{\partial \delta^2} + \lambda^* \frac{\partial^2 L(\delta^*, \theta)}{\partial \delta^2} & \frac{\partial L(\delta^*, \theta)}{\partial \delta} \\ \frac{\partial L(\delta^*, \theta)}{\partial \delta}^\top & 0 \end{pmatrix}$

*is full rank, then*

$$\nabla_\theta d_\theta(x, y) = C(\theta, x, y) \frac{\partial L(\delta^*, \theta)}{\partial \theta}, \ where \ C(\theta, x, y) = \frac{\left\langle \frac{\partial \epsilon(\delta^*)}{\partial \delta}, \frac{\partial L(\delta^*, \theta)}{\partial \delta} \right\rangle}{\|\frac{\partial L(\delta^*, \theta)}{\partial \delta}\|_2^2} \ is \ a \ scalar.$$

**Remark 2.1.** *By Proposition 2.1, the margin's gradient w.r.t. to the model parameter $\theta$ is proportional to the loss' gradient w.r.t. $\theta$ at $\delta^*$, the shortest successful perturbation. Therefore to perform gradient ascent on the margin, we just need to find $\delta^*$ and perform gradient descent on the loss.*

**Margin maximization for non-smooth loss and norm:** Proposition 2.1 requires the loss function and the norm to be $C^2$ at $\delta^*$. This might not be the case for many functions used in practice, e.g. ReLU networks and the $\ell_\infty$ norm. Our next result shows that under a weaker condition of directional differentiability (instead of $C^2$), learning $\theta$ to maximize the margin can still be done by decreasing $L(\delta^*, \theta)$ w.r.t. $\theta$, at $\theta = \theta_0$. Due to space limitations, we only present an informal statement here. Rigorous statements can be found in Appendix A.2.

**Proposition 2.2.** *Let $\delta^*$ be unique and $L(\delta, \theta)$ be the loss of a deep ReLU network. There exists some direction $\vec{v}$ in the parameter space, such that the loss $L(\delta, \theta)|_{\delta = \delta^*}$ can be reduced in the direction of $\vec{v}$. Furthermore, by reducing $L(\delta, \theta)|_{\delta = \delta^*}$, the margin is also guaranteed to be increased.*

Figure 2 illustrates the relationship between the margin and the adversarial loss with an imaginary example. Consider a 1-D example in Figure 2 (a), where the input example $x$ is a scalar. We perturb $x$ in the positive direction with perturbation $\delta$. As we fix $(x, y)$, we overload $L(\delta, \theta) = L_\theta^{\mathrm{LM}}(x + \delta, y)$, which is monotonically increasing on $\delta$, namely larger perturbation results in higher loss. Let $L(\cdot, \theta_0)$ (the dashed curve) denote the original function before an update step, and $\delta_0^* = \arg\min_{L(\delta, \theta_0) \geq 0} \|\delta\|$ denote the corresponding margin (same as shortest successful perturbation in

1D). As shown in Figure 2 (b), as the parameter is updated to $\theta_1$ such that $L(\delta_0^*, \theta_1)$ is reduced, the new margin $\delta_1^* = \arg\min_{L(\delta, \theta_1) \geq 0} \|\delta\|$ is enlarged. Intuitively, a reduced value of the loss at the shortest successful perturbation leads to an increase in margin.

## 2.2 STABILIZING THE LEARNING WITH CROSS ENTROPY SURROGATE LOSS

In practice, we find the gradients of the "logit margin loss" $L_\theta^{\text{LM}}$ to be unstable. The piecewise nature of the $L_\theta^{\text{LM}}$ loss can lead to discontinuity of its gradient, causing large fluctuations on the boundary between the pieces. It also does not fully utilize information provided by all the logits.

In our MMA algorithm, we instead use the *"soft logit margin loss"* (SLM)

$$L_\theta^{\text{SLM}}(x, y) = \log \sum_{j \neq y} \exp(f_\theta^j(x)) - f_\theta^y(x),$$

which serves as a surrogate loss to the *"logit margin loss"* $L_\theta^{\text{LM}}(x, y)$ by replacing the the $\max$ function by the LogSumExp (sometimes also called softmax) function. One immediate property is that the SLM loss is smooth and convex (w.r.t. logits). The next proposition shows that SLM loss is a good approximation to the LM loss.

**Proposition 2.3.**
$$L_\theta^{SLM}(x, y) - \log(K - 1) \leq L_\theta^{LM}(x, y) \leq L_\theta^{SLM}(x, y), \tag{4}$$
*where $K$ denote the number of classes.*

**Remark 2.2.** *By using the soft logit margin loss, MMA maximizes a lower bound of the margin, the SLM-margin, $d_\theta^{SLM}(x, y)$:*

$$d_\theta^{SLM}(x, y) = \|\delta^*\| = \min \|\delta\| \quad s.t. \ \delta : L_\theta^{SLM}(x + \delta, y) \geq 0.$$

*To see that, note by Proposition 2.3, $L_\theta^{SLM}(x, y)$ upper bounds $L_\theta^{LM}(x, y)$. So we have $\{\delta : L_\theta^{LM}(x + \delta, y) \leq 0\} \subseteq \{\delta : L_\theta^{SLM}(x + \delta, y) \leq 0\}$. Therefore, $d_\theta^{SLM}(x, y) \leq d_\theta(x, y)$, i.e. the SLM-margin is a lower bound of the margin.*

We next show the gradient of the SLM loss is proportional to the gradient of the cross entropy loss, $L_\theta^{\text{CE}}(x, y) = \log \sum_j \exp(f_\theta^j(x)) - f_\theta^y(x)$, thus minimizing $L_\theta^{\text{CE}}(x + \delta^*, y)$ w.r.t. $\theta$ "is" minimizing $L_\theta^{\text{SLM}}(x + \delta^*, y)$.

**Proposition 2.4.** *For a fixed $(x, y)$ and $\theta$,*

$$\nabla_\theta L_\theta^{CE}(x, y) = r(\theta, x, y)\nabla_\theta L_\theta^{SLM}(x, y), \text{ and } \nabla_x L_\theta^{CE}(x, y) = r(\theta, x, y)\nabla_x L_\theta^{SLM}(x, y) \tag{5}$$

$$\text{where the scalar } r(\theta, x, y) = \frac{\sum_{i \neq y} \exp(f_\theta^i(x))}{\sum_i \exp(f_\theta^i(x))}. \tag{6}$$

Therefore, to simplify the learning algorithm, we perform gradient descent on model parameters using $L_\theta^{\text{CE}}(x + \delta^*, y)$. As such, we use $L_\theta^{\text{CE}}$ on both clean and adversarial examples, which in practice stabilizes training:

$$\min_\theta L_\theta^{\text{MMA}}(\mathcal{S}), \text{ where } L_\theta^{\text{MMA}}(\mathcal{S}) = \sum_{i \in \mathcal{S}_\theta^+ \cap \mathcal{H}_\theta} L_\theta^{\text{CE}}(x_i + \delta^*, y_i) + \sum_{j \in \mathcal{S}_\theta^-} L_\theta^{\text{CE}}(x_j, y_j), \tag{7}$$

where $\delta^* = \arg\min_{L_\theta^{\text{SLM}}(x+\delta, y) \geq 0} \|\delta\|$ is found with the SLM loss, and $\mathcal{H}_\theta = \{i : d_\theta(x_i, y_i) < d_{\max}\}$ is the set of examples that have margins smaller than the hinge threshold.

## 2.3 FINDING THE OPTIMAL PERTURBATION $\delta^*$

To implement MMA, we still need to find the $\delta^*$, which is intractable in general settings. We propose an adaptation of the projected gradient descent (PGD) (Madry et al., 2017) attack to give an approximate solution of $\delta^*$, the Adaptive Norm Projective Gradient Descent Attack (AN-PGD). In AN-PGD, we apply PGD on an initial perturbation magnitude $\epsilon_{init}$ to find a norm-constrained perturbation $\delta_1$, then we search along the direction of $\delta_1$ to find a scaled perturbation that gives $L = 0$, we then use this scaled perturbation to approximate $\epsilon^*$. Note that AN-PGD here only serves as an algorithm to give an approximate solution of $\delta^*$, and it can be decoupled from the remaining parts of MMA training. Other attacks that can serve a similar purpose can also fit into our MMA training framework, e.g. the Decoupled Direction and Norm (DDN) attack (Rony et al., 2018). Algorithm 1 describes the Adaptive Norm PGD Attack (AN-PGD) algorithm.

---

**Algorithm 1** Adaptive Norm PGD Attack for approximately solving $\delta^*$.

**Inputs**: $(x, y)$ is the data example. $\epsilon_{init}$ is the initial norm constraint used in the first PGD attack.
**Outputs**: $\delta^*$, approximate shortest successful perturbation. **Parameters**: $\epsilon_{max}$ is the maximum perturbation length. $\mathrm{PGD}(x, y, \epsilon)$ represents PGD perturbation $\delta$ with magnitude $\epsilon$.

---

1: Adversarial example $\delta_1 = \mathrm{PGD}(x, y, \epsilon_{init})$
2: Unit perturbation $\delta_u = \frac{\delta_1}{\|\delta_1\|}$
3: **if** prediction on $x + \delta_1$ is correct **then**
4:     Bisection search to find $\epsilon'$, the zero-crossing of $L(x + \eta\delta_u, y)$ w.r.t. $\eta, \eta \in [\|\delta_1\|, \epsilon_{max}]$
5: **else**
6:     Bisection search to find $\epsilon'$, the zero-crossing of $L(x + \eta\delta_u, y)$ w.r.t. $\eta, \eta \in [0, \|\delta_1\|)$
7: **end if**
8: $\delta^* = \epsilon'\delta_u$

---

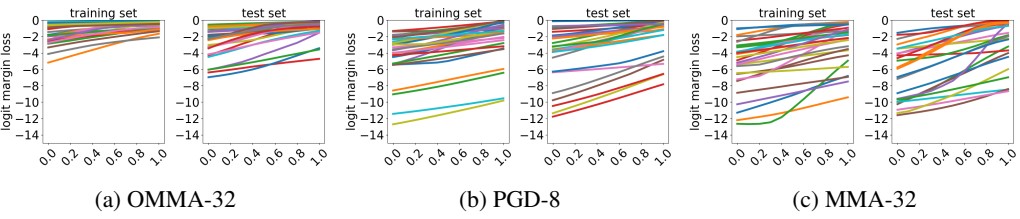

Figure 3: Visualization of loss landscape in the input space for MMA and PGD trained models.

**Remark 2.3.** *Finding the $\delta^*$ in Propositions 2.1 and 2.2 requires solving a non-convex optimization problem, where the optimality cannot be guaranteed in practice. Previous adversarial training methods, e.g. Madry et al. (2017), suffer the same problem. Nevertheless, as we show later in Figure 4, our proposed MMA training algorithm does achieve the desired behavior of maximizing the margin of each individual example in practice.*

### 2.4 ADDITIONAL CLEAN LOSS DURING TRAINING

In practice, we observe that when the model is only trained with the objective function in Eq. (7), the input space loss landscape is very flat, which makes PGD less efficient in finding $\delta^*$ for training, as shown in Figure 3. Here we choose 50 examples from both the training and test sets respectively, then perform the PGD attack with $\epsilon = 8/255$ and keep those failed perturbations. For each example, we linearly interpolate 9 more points between the original example and the perturbed example, and plot their logit margin losses. In each sub-figure, the horizontal axis is the relative position of the interpolated example: e.g. 0.0 represents the original example, 1.0 represents the perturbed example with $\epsilon = 8/255$, 0.5 represents the average of them. The vertical axis is the logit margin loss. Recall that when $L_\theta^{\mathrm{LM}}(x + \delta, y) < 0$, the perturbation $\delta$ fails.

OMMA-32 in Figure 3a represents model trained with only $L_\theta^{\mathrm{MMA}}$ in Eq. (7) with $d_{\max} = 8$. PGD-8 Figure 3b represents model trained with PGD training (Madry et al., 2017) with $\epsilon = 8$. As one can see, OMMA-32 has "flatter" loss curves compared to PGD-8. This could potentially weaken the adversary during training, which leads to poor approximation of $\delta^*$ and hampers training.

To alleviate this issue, we add an additional clean loss term to the MMA objective in Eq. (7) to lower the loss on clean examples, such that the input space loss landscape is steeper. Specifically, we use the following combined loss

$$L_\theta^{\mathrm{CB}}(\mathcal{S}) = \frac{1}{3} \sum_{j \in \mathcal{S}} L_\theta^{\mathrm{CE}}(x_j, y_j) + \frac{2}{3} L_\theta^{\mathrm{MMA}}(\mathcal{S}). \tag{8}$$

The model trained with this combined loss and $d_{\max} = 32$ is the MMA-32 shown in Figure 3c. Adding the clean loss is indeed effective. Most of the loss curves are more tilted, and the losses of perturbed examples are lower. We use $L_\theta^{\mathrm{CB}}$ for MMA training in the rest of the paper due to its higher performance. A more detailed comparison between $L_\theta^{\mathrm{CB}}$ and $L_\theta^{\mathrm{MMA}}$ is delayed to Appendix E.

### 2.5 THE MMA TRAINING ALGORITHM

---

**Algorithm 2** Max-Margin Adversarial Training.
**Inputs**: The training set $\{(x_i, y_i)\}$. **Outputs**: the trained model $f_\theta(\cdot)$. **Parameters**: $\epsilon$ contains perturbation lengths of training data. $\epsilon_{min}$ is the minimum perturbation length. $\epsilon_{max}$ is the maximum perturbation length. $\mathcal{A}(x, y, \epsilon_{init})$ represents the approximate shortest successful perturbation returned by an algorithm $\mathcal{A}$ (e.g. AN-PGD) on the data example $(x, y)$ and at the initial norm $\epsilon_{init}$.

---

1: Randomly initialize the parameter $\theta$ of model $f$, and initialize every element of $\epsilon$ as $\epsilon_{min}$
2: **repeat**
3:     Read minibatch $B = \{(x_1, y_1), \ldots, (x_m, y_m)\}$
4:     Make predictions on $B$ and into two: wrongly predicted $B_0$ and correctly predicted $B_1$
5:     Initialize an empty batch $B_1^{\text{adv}}$
6:     **for** $(x_i, y_i)$ in $B_1$ **do**
7:         Retrieve perturbation length $\epsilon_i$ from $\epsilon$
8:         $\delta_i^* = \mathcal{A}(x_i, y_i, \epsilon_i)$
9:         Update the $\epsilon_i$ in $\epsilon$ as $\|\delta_i^*\|$. If $\|\delta_i^*\| < d_{\max}$ then put $(x_i + \delta_i^*, y_i)$ into $B_1^{\text{adv}}$
10:     **end for**
11:     Calculate gradients of $\sum_{j \in B^0} L_\theta^{\text{CE}}(x_j, y_j) + \frac{1}{3} \sum_{j \in B^1} L_\theta^{\text{CE}}(x_j, y_j) + \frac{2}{3} \sum_{j \in B_1^{\text{adv}}} L_\theta^{\text{CE}}(x_j, y_j)$, the combined loss, on $B_0$, $B_1$, and $B_1^{\text{adv}}$, w.r.t. $\theta$, according to Eqs. (7) and (8)
12:     Perform one step gradient step update on $\theta$
13: **until** meet training stopping criterion

---

Algorithm 2 summarizes our practical MMA training algorithm. During training for each minibatch, we 1) separate it into 2 batches based on if the current prediction matches the label; 2) find $\delta^*$ for each example in the "correct batch"; 3) calculate the gradient of $\theta$ based on Eqs. (7) and (8).

## 3 UNDERSTANDING ADVERSARIAL TRAINING W.R.T. MARGIN

Through our development of MMA training in the last section, we have shown that margin maximization is closely related to adversarial training with the optimal perturbation length $\|\delta^*\|$. In this section, we further investigate the behavior of adversarial training in the perspective of margin maximization. Adversarial training (Huang et al., 2015; Madry et al., 2017) minimizes the "worst-case" loss under a fixed perturbation magnitude $\epsilon$, as follows.

$$\min_\theta \mathbb{E}_{x, y \sim \mathcal{D}} \max_{\|\delta\| \leq \epsilon} L_\theta(x + \delta, y). \tag{9}$$

Looking again at Figure 2, we can see that an adversarial training update step does not necessarily increase the margin. In particular, as we perform an update to reduce the value of loss at the fixed perturbation $\epsilon$, the parameter is updated from $\theta_0$ to $\theta_1$. After this update, we can imagine two different scenarios for the updated loss functions $L_{\theta_1}(\cdot)$ (the solid curve) in Figure 2 (c) and (d). In both (c) and (d), $L_{\theta_1}(\epsilon)$ is decreased by the same amount. However, the margin is increased in (c) with $\delta_1^* > \delta_0^*$, but decreased in (d) with $\delta_1^* < \delta_0^*$. Formalizing the intuitive analysis, we present two theorems connecting adversarial training and margin maximization. For brevity, fixing $(x, y)$, let $L(\delta, \theta) = L_\theta^{\text{LM}}(x + \delta, y)$, $d_\theta = d_\theta(x, y)$, and $\epsilon_\theta^*(\rho) = \min_{\delta : L(\delta, \theta) \geq \rho} \|\delta\|$.

**Theorem 3.1.** *Assuming an update from adversarial training changes $\theta_0$ to $\theta_1$, such that $\rho^* = \max_{\|\delta\| \leq \epsilon} L(\delta, \theta_0) > \max_{\|\delta\| \leq \epsilon} L(\delta, \theta_1)$, then*

*1) if $\epsilon = d_{\theta_0}$, then $\rho^* = 0$, $\epsilon_{\theta_1}^*(\rho^*) = d_{\theta_1} \geq d_{\theta_0} = \epsilon_{\theta_0}^*(\rho^*)$;*

*2) if $\epsilon < d_{\theta_0}$, then $\rho^* \leq 0$, $\epsilon_{\theta_0}^*(\rho^*) \leq d_{\theta_0}$, $\epsilon_{\theta_1}^*(\rho^*) \leq d_{\theta_1}$, and $\epsilon_{\theta_1}^*(\rho^*) \geq \epsilon_{\theta_0}^*(\rho^*)$;*

*3) if $\epsilon > d_{\theta_0}$, then $\rho^* \geq 0$, $\epsilon_{\theta_0}^*(\rho^*) \geq d_{\theta_0}$, $\epsilon_{\theta_1}^*(\rho^*) \geq d_{\theta_1}$, and $\epsilon_{\theta_1}^*(\rho^*) \geq \epsilon_{\theta_0}^*(\rho^*)$.*

**Remark 3.1.** *In other words, adversarial training, with the logit margin loss and a fixed $\epsilon$*

*1) exactly maximizes the margin, if $\epsilon$ is equal to the margin;*

*2) maximizes a lower bound of the margin, if $\epsilon$ is less than the margin;*

*3) maximizes an upper bound of the margin, if $\epsilon$ is greater than the margin.*

Next we look at adversarial training with the cross entropy loss through the connection between cross entropy and the soft logit margin loss from Proposition 2.4. We first look at adversarial training on the SLM loss. Fixing $(x, y)$, let $d_\theta^{\text{SLM}} = d_\theta^{\text{SLM}}(x, y)$, and $\epsilon_{\text{SLM}, \theta}^*(\rho) = \min_{L_\theta^{\text{SLM}}(x + \delta, y) \geq \rho} \|\delta\|$.

**Corollary 3.1.** *Assuming an update from adversarial training changes $\theta_0$ to $\theta_1$, such that $\max_{\|\delta\| \leq \epsilon} L_{\theta_0}^{SLM}(x + \delta, y) > \max_{\|\delta\| \leq \epsilon} L_{\theta_1}^{SLM}(x + \delta, y)$, if $\epsilon \leq d_{\theta_0}^{SLM}$, then $\rho^* = \max_{\|\delta\| \leq \epsilon} L_{\theta_0}^{SLM}(x + \delta, y) \leq 0$, $\epsilon_{SLM, \theta_0}^*(\rho^*) \leq d_{\theta_0}$, $\epsilon_{SLM, \theta_1}^*(\rho^*) \leq d_{\theta_1}$, and $\epsilon_{SLM, \theta_1}^*(\rho^*) \geq \epsilon_{SLM, \theta_0}^*(\rho^*)$.*

**Remark 3.2.** *In other words, if $\epsilon$ is less than or equal to the SLM-margin, adversarial training, with the SLM loss and a fixed perturbation length $\epsilon$, maximizes a lower bound of the SLM-margin, thus a lower bound of the margin.*

Recall Proposition 2.4 shows that $L_\theta^{\text{CE}}$ and $L_\theta^{\text{SLM}}$ have the same gradient direction w.r.t. both the model parameter and the input. In adversarial training (Madry et al., 2017), the PGD attack only uses the gradient direction w.r.t. the input, but not the gradient magnitude. Therefore, in the inner maximization loop, using the SLM and CE loss will result in the same approximate $\delta^*$. Furthermore, $\nabla_\theta L_\theta^{\text{CE}}(x + \delta^*, y)$ and $\nabla_\theta L_\theta^{\text{SLM}}(x + \delta^*, y)$ have the same direction. If the step size is chosen appropriately, then a gradient update that reduces $L_\theta^{\text{CE}}(x + \delta^*, y)$ will also reduce $L_\theta^{\text{SLM}}(x + \delta^*, y)$. Combined with Remark 3.2, these suggest:

*Adversarial training with cross entropy loss (Madry et al., 2017) approximately maximizes a lower bound of the margin, if $\epsilon$ is smaller than or equal to the SLM-margin.*

On the other hand, when $\epsilon$ is larger then the margin, such a relation no longer exists. When $\epsilon$ is too large, adversarial training is maximizing an upper bound of the margin, which might not necessarily increase the margin. This suggests that for adversarial training with a large $\epsilon$, starting with a smaller $\epsilon$ then gradually increasing it could help, since the lower bound of the margin is maximized at the start of training. Results in Sections 4.1 and 4.2 corroborate exactly with this theoretical prediction.

## 4 EXPERIMENTS

We empirically examine several hypotheses and compare MMA training with different adversarial training algorithms on the MNIST and CIFAR10 datasets under $\ell_\infty/\ell_2$-norm constrained perturbations. Due to space limitations, we mainly present results on CIFAR10-$\ell_\infty$ for representative models in Table 1. Full results are in Table 2 to 15 in Appendix F. Implementation details are also left to the appendix, including neural network architecture, training and attack hyperparameters.

Our results confirm our theory and show that MMA training is stable to its hyperparameter $d_{\max}$, and balances better among various attack lengths compared to adversarial training with fixed perturbation magnitude. This suggests that MMA training is a better choice for defense when the perturbation length is unknown, which is often the case in practice.

**Measuring Adversarial Robustness:** We use the robust accuracy under multiple projected gradient descent (PGD) attacks (Madry et al., 2017) as the robustness measure. Specifically, given an example, each model is attacked by both repeated randomly initialized whitebox PGD attacks and numerous transfer attacks, generated from whitebox PGD attacking other models. If any one of these attacks succeed, then the model is considered "not robust under attack" on this example. For each dataset-norm setting and for each example, under a particular magnitude $\epsilon$, we first perform $N$ randomly initialized whitebox PGD attacks on each individual model, then use $N \cdot (m - 1)$ PGD attacks from all the other models to perform transfer attacks, where $m$ is the total number of models considered under each setting (e.g. CIFAR10-$\ell_\infty$). In our experiments, we use $N = 10$ for models trained on CIFAR10, thus the total number of the "combined" (whitebox and transfer) set of attacks is 320 for CIFAR10-$\ell_\infty$ ($m = 32$). [3] We use *ClnAcc* for clean accuracy, *AvgAcc* for the average over both clean accuracy and robust accuracies at different $\epsilon$'s, *AvgRobAcc* for the average over only robust accuracies under attack.

**Correspondence between $d_{\max}$ and $\epsilon$:** In MMA training, we maximize each example's margin until the margin reaches $d_{\max}$. In standard adversarial training, each example is trained to be robust at $\epsilon$. Therefore in MMA training, we usually set $d_{\max}$ to be larger than $\epsilon$ in standard adversarial training. It is difficult to know the "correct" value of $d_{\max}$, therefore we test different $d_{\max}$ values, and compare a group of MMA trained models to a group of PGD trained models.

### 4.1 EFFECTIVENESS OF MARGIN MAXIMIZATION DURING TRAINING

As discussed in Section 3, MMA training enlarges margins of all training points,while PGD training, by minimizing the adversarial loss with a fixed $\epsilon$, might fail to enlarge margins for points with initial margins smaller than $\epsilon$. This is because when $d_\theta(x, y) < \epsilon$, PGD training is maximizing an upper bound of $d_\theta(x, y)$, which may not necessarily increase the margin. To verify this, we track how the

---

[3] $N = 50$ for models trained on MNIST. Total number of attacks is 900 for MNIST-$\ell_\infty$ ($m = 18$), 1200 for MNIST-$\ell_2$ ($m = 24$) and 260 for CIFAR10-$\ell_2$ ($m = 26$). We explain relevant details in Appendix C.

Figure 4: Margin distributions during training, under the CIFAR10-$\ell_2$ case. Each blue histogram represents the margin value distribution of MMA-3.0, and the orange represents PGD-2.5.

Table 1: Accuracies of representative models trained on CIFAR10 with $\ell_\infty$-norm constrained attacks. Robust accuracies are calculated under combined (whitebox+transfer) PGD attacks. AvgAcc averages over clean and all robust accuracies; AvgRobAcc averages over all robust accuracies.

| CIFAR10 Model | Cln Acc | AvgAcc | AvgRobAcc | RobAcc under different $\epsilon$, combined (whitebox+transfer) attacks | | | | | | | |
|---|---|---|---|---|---|---|---|---|---|---|---|
| | | | | 4 | 8 | 12 | 16 | 20 | 24 | 28 | 32 |
| PGD-8 | 85.14 | 27.27 | 20.03 | 67.73 | 46.47 | 26.63 | 12.33 | 4.69 | 1.56 | 0.62 | 0.22 |
| PGD-16 | 68.86 | 28.28 | 23.21 | 57.99 | 46.09 | 33.64 | 22.73 | 13.37 | 7.01 | 3.32 | 1.54 |
| PGD-24 | 10.90 | 9.95 | 9.83 | 10.60 | 10.34 | 10.11 | 10.00 | 9.89 | 9.69 | 9.34 | 8.68 |
| PGDLS-8 | 85.63 | 27.20 | 19.90 | 67.96 | 46.19 | 26.19 | 12.22 | 4.51 | 1.48 | 0.44 | 0.21 |
| PGDLS-16 | 70.68 | 28.44 | 23.16 | 59.43 | 47.00 | 33.64 | 21.72 | 12.66 | 6.54 | 2.98 | 1.31 |
| PGDLS-24 | 58.36 | 26.53 | 22.55 | 49.05 | 41.13 | 32.10 | 23.76 | 15.70 | 9.66 | 5.86 | 3.11 |
| MMA-12 | 88.59 | 26.87 | 19.15 | 67.96 | 43.42 | 24.07 | 11.45 | 4.27 | 1.43 | 0.45 | 0.16 |
| MMA-20 | 86.56 | 28.86 | 21.65 | 66.92 | 46.89 | 29.83 | 16.55 | 8.14 | 3.25 | 1.17 | 0.43 |
| MMA-32 | 84.36 | 29.39 | 22.51 | 64.82 | 47.18 | 31.49 | 18.91 | 10.16 | 4.77 | 1.97 | 0.81 |
| PGD-ens | 87.38 | 28.10 | 20.69 | 64.59 | 46.95 | 28.88 | 15.10 | 6.35 | 2.35 | 0.91 | 0.39 |
| PGDLS-ens | 76.73 | 29.52 | 23.62 | 60.52 | 48.21 | 35.06 | 22.14 | 12.28 | 6.17 | 3.14 | 1.43 |
| TRADES | 84.92 | 30.46 | 23.65 | 70.96 | 52.92 | 33.04 | 18.23 | 8.34 | 3.57 | 1.4 | 0.69 |

margin distribution changes during training processes in two models under the CIFAR10-$\ell_2$ [4] case, MMA-3.0 vs PGD-2.5. We use *MMA-$d_{\max}$* to denote the MMA trained model with the combined loss in Eq. (8) and hinge threshold $d_{\max}$, and *PGD-$\epsilon$* to represent the PGD trained (Madry et al., 2017) model with fixed perturbation magnitude $\epsilon$.

Specifically, we randomly select 500 training points, and measure their margins after each training epoch. We use the norm of the 1000-step DDN attack (Rony et al., 2018) to approximate the margin. The results are shown in Figure 4, where each subplot is a histogram (rotated by 90°) of margin values. For the convenience of comparing across epochs, we use the vertical axis to indicate margin value, and the horizontal axis for counts in the histogram. The number below each subplot is the corresponding training epoch. Margins mostly concentrate near 0 for both models at the beginning. As training progresses, both enlarge margins on average. However, in PGD training, a portion of the margins stay close to 0 throughout the training process, at the same time, also pushing some margins to be even higher than 2.5, likely because PGD training continues to maximize lower bounds of this subset of the total margins, as we discussed in Section 3, the $\epsilon$ value that the PGD-2.5 model is trained for. MMA training, on the other hand, does not "give up" on those data points with small margins. At the end of training, 37.8% of the data points for PGD-2.5 have margins smaller than 0.05, while only 20.4% for MMA. As such, PGD training enlarges the margins of "easy data" which are already robust enough, but "gives up" on "hard data" with small margins. Instead, MMA training pushes the margin of every data point, by finding the proper $\epsilon$. In general, when the attack magnitude is unknown, MMA training is more capable in achieving a better balance between small and large margins, and thus a better balance among adversarial attacks with various $\epsilon$'s as a whole.

## 4.2 GRADUALLY INCREASING $\epsilon$ HELPS PGD TRAINING WHEN $\epsilon$ IS LARGE

Our previous analysis in Section 3 suggests that when the fixed perturbation magnitude $\epsilon$ is small, PGD training increases the lower bound of the margin. On the other hand, when $\epsilon$ is larger than the margin, PGD training does not necessarily increase the margin. This is indeed confirmed by our experiments. PGD training fails at larger $\epsilon$, in particular $\epsilon = 24/255$ for the CIFAR10-$\ell_\infty$ as shown in Table 1. We can see that PGD-24's accuracies at all test $\epsilon$'s are around 10%.

---

[4] Here we choose CIFAR10-$\ell_2$ because the DDN-$\ell_2$ attack is both fast and effective for estimating margins.

Aiming to improve PGD training, we propose a variant of PGD training, named PGD with Linear Scaling (PGDLS), in which we grow the perturbation magnitude from 0 to the fixed magnitude linearly in 50 epochs. According to our theory, gradually increasing the perturbation magnitude could avoid picking a $\epsilon$ that is larger than the margin, thus managing to maximizing the lower bound of the margin rather than its upper bound, which is more sensible. It can also be seen as a "global magnitude scheduling" shared by all data points, which is to be contrasted to MMA training that gives magnitude scheduling for each individual example. We use *PGDLS-$\epsilon$* to represent these models and show their performances also in Table 1. We can see that PGDLS-24 is trained successfully, whereas PGD-24 fails. At $\epsilon = 8$ or 16, PGDLS also performs similar or better than PGD training, confirming the benefit of training with small perturbation at the beginning.

### 4.3 COMPARING MMA TRAINING WITH PGD TRAINING

From the first three columns in Table 1, we can see that MMA training is very stable with respect to its hyperparameter, the hinge threshold $d_{\max}$. When $d_{\max}$ is set to smaller values (e.g. 12 and 20), MMA models attain good robustness across different attacking magnitudes, with the best clean accuracies in the comparison set. When $d_{\max}$ is large, MMA training can still learn a reasonable model that is both accurate and robust. For MMA-32, although $d_{\max}$ is set to a "impossible-to-be-robust" level at 32/255, it still achieves 84.36% clean accuracy and 47.18% robust accuracy at 8/255, thus automatically "ignoring" the demand to be robust at larger $\epsilon$'s, including 20, 24, 28 and 32, recognizing its infeasibility due to the intrinsic difficulty of the problem. In contrast, PGD trained models are more sensitive to their specified fixed perturbation magnitude. In terms of the overall performance, we notice that MMA training with a large $d_{\max}$, e.g. 20 or 32, achieves high AvgAcc values, e.g. 28.86% or 29.39%. However, for PGD training to achieve a similar performance, $\epsilon$ needs to be carefully picked (PGD-16 and PGDLS-16), and their clean accuracies suffer a significant drop.

We also compare MMA models with ensembles of PGD trained models. *PGD-ens/PGDLS-ens* represents the ensemble of PGD/PGDLS trained models with different $\epsilon$'s. The ensemble produces a prediction by performing a majority vote on label predictions, and using the softmax scores as the tie breaker. MMA training achieves similar performance compared to the ensembled PGD models. PGD-ens maintains a good clean accuracy, but it is still marginally outperformed by MMA-32 w.r.t. robustness at various $\epsilon$'s. Further note that 1) the ensembling requires significantly higher computation costs both at training and test times; 2) Unlike attacking individual models, attacking ensembles is still relatively unexplored in the literature, thus our whitebox PGD attacks on the ensembles may not be sufficiently effective; [5] and 3) as shown in Appendix F, for MNIST-$\ell_\infty/\ell_2$, MMA trained models significantly outperform the PGD ensemble models.

**Comparison with TRADES:** TRADES (Zhang et al., 2019b) improves PGD training by decoupling the minimax training objective into an accuracy term and an robustness regularization term. We compare MMA-32 with TRADES [6] in Table 1 as their clean accuracies are similar. We see that TRADES outperforms MMA on the attacks of lengths that are less than $\epsilon = 12/255$, but it sacrifices the robustness under larger attacks. We note that MMA training and TRADES' idea of optimizing a calibration loss are progresses in orthogonal directions and could potentially be combined.

**Testing on gradient free attacks:** As a sanity check for gradient obfuscation (Athalye et al., 2018), we also performed the gradient-free SPSA attack (Uesato et al., 2018), to all our $\ell_\infty$-MMA trained models on the first 100 test examples. We find that, in all cases, SPSA does not compute adversarial examples successfully when gradient-based PGD did not.

## 5 CONCLUSIONS

In this paper, we proposed to directly maximize the margins to improve adversarial robustness. We developed the MMA training algorithm that optimizes the margins via adversarial training with perturbation magnitude adapted both throughout training and individually for the distinct datapoints in the training dataset. Furthermore, we rigorously analyzed the relation between adversarial training and margin maximization. Our experiments on CIFAR10 and MNIST empirically confirmed our theory and demonstrate that MMA training outperforms adversarial training in terms of sensitivity to hyperparameter setting and robustness to variable attack lengths, suggesting MMA is a better choice for defense when the adversary is unknown, which is often the case in practice.

---

[5]Attacks on ensembles are explained in Appendix D.

[6]Downloaded models from https://github.com/yaodongyu/TRADES

ACKNOWLEDGEMENT

We thank Marcus Brubaker, Bo Xin, Zhanxing Zhu, Joey Bose and Hamidreza Saghir for proofreading earlier drafts and useful feedbacks, also Jin Sung Kang for helping with computation resources.

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

# Appendix

## A   PROOFS

### A.1   PROOF OF PROPOSITION 2.1

*Proof.* Recall $\epsilon(\delta) = \|\delta\|$. Here we compute the gradient for $d_\theta(x, y)$ in its general form. Consider the following optimization problem:

$$d_\theta(x, y) = \min_{\delta \in \Delta(\theta)} \epsilon(\delta),$$

where $\Delta(\theta) = \{\delta : L_\theta(x+\delta, y) = 0\}$, $\epsilon$ and $L(\delta, \theta)$ are both $C^2$ functions [7]. Denotes its Lagrangian by $\mathcal{L}(\delta, \lambda)$, where

$$\mathcal{L}(\delta, \lambda) = \epsilon(\delta) + \lambda L_\theta(x + \delta, y)$$

For a fixed $\theta$, the optimizer $\delta^*$ and $\lambda^*$ must satisfy the first-order conditions (FOC)

$$\frac{\partial \epsilon(\delta)}{\partial \delta} + \lambda \frac{\partial L_\theta(x + \delta, y)}{\partial \delta}\bigg|_{\delta = \delta^*, \lambda = \lambda^*} = 0, \tag{10}$$

$$L_\theta(x + \delta, y)|_{\delta = \delta^*} = 0.$$

Put the FOC equations in vector form,

$$G((\delta, \lambda), \theta) = \left( \begin{array}{c} \frac{\partial \epsilon(\delta)}{\partial \delta} + \lambda \frac{\partial L_\theta(x+\delta, y)}{\partial \delta} \\ L_\theta(x + \delta, y) \end{array} \right)\bigg|_{\delta = \delta^*, \lambda = \lambda^*} = 0.$$

Note that $G$ is $C^1$ continuously differentiable since $\epsilon$ and $L(\delta, \theta)$ are $C^2$ functions. Furthermore, the Jacobian matrix of $G$ w.r.t $(\delta, \lambda)$ is

$$\nabla_{(\delta, \lambda)} G((\delta^*, \lambda^*), \theta)$$

$$= \left( \begin{array}{cc} \frac{\partial^2 \epsilon(\delta^*)}{\partial \delta^2} + \lambda^* \frac{\partial^2 L(\delta^*, L(\delta, \theta))}{\partial \delta^2} & \frac{\partial L(\delta^*, \theta)}{\partial \delta} \\ \frac{\partial L(\delta^*, \theta)}{\partial \delta}^\top & 0 \end{array} \right)$$

which by assumption is full rank. Therefore, by the implicit function theorem, $\delta^*$ and $\lambda^*$ can be expressed as a function of $\theta$, denoted by $\delta^*(\theta)$ and $\lambda^*(\theta)$.

To further compute $\nabla_\theta d_\theta(x, y)$, note that $d_\theta(x, y) = \epsilon(\delta^*(\theta))$. Thus,

$$\nabla_\theta d_\theta(x, y) = \frac{\partial \epsilon(\delta^*)}{\partial \delta} \frac{\partial \delta^*(\theta)}{\partial \theta} = -\lambda^* \frac{\partial L(\delta^*, \theta)}{\partial \delta} \frac{\partial \delta^*(\theta)}{\partial \theta}, \tag{11}$$

where the second equality is by Eq. (10). The implicit function theorem also provides a way of computing $\frac{\partial \delta^*(\theta)}{\partial \theta}$ which is complicated involving taking inverse of the matrix $\nabla_{(\delta, \lambda)} G((\delta^*, \lambda^*), \theta)$.

Here we present a relatively simple way to compute this gradient. Note that by the definition of $\delta^*(\theta)$,

$$L(\delta^*(\theta), \theta) \equiv 0.$$

and $\delta^*(\theta)$ is a differentiable implicit function of $\theta$ restricted to this level set. Differentiate with w.r.t. $\theta$ on both sides:

$$\frac{\partial L(\delta^*, \theta)}{\partial \theta} + \frac{\partial L(\delta^*, \theta)}{\partial \delta} \frac{\partial \delta^*(\theta)}{\partial \theta} = 0. \tag{12}$$

Combining Eq. (11) and Eq. (12),

$$\nabla_\theta d_\theta(x, y) = \lambda^*(\theta) \frac{\partial L(\delta^*, \theta)}{\partial \theta}. \tag{13}$$

Lastly, note that

$$\left\| \frac{\partial \epsilon(\delta)}{\partial \delta} + \lambda \frac{\partial L_\theta(x + \delta, y)}{\partial \delta}\bigg|_{\delta = \delta^*, \lambda = \lambda^*} \right\|_2 = 0.$$

---

[7]Note that a simple application of Danskin's theorem would not be valid as the constraint set $\Delta(\theta)$ depends on the parameter $\theta$.

Therefore, one way to calculate $\lambda^*(\theta)$ is by

$$\lambda^*(\theta) = \frac{\frac{\partial \epsilon(\delta)}{\partial \delta}^\top \frac{\partial L_\theta(x+\delta,y)}{\partial \delta}}{\frac{\partial L_\theta(x+\delta,y)}{\partial \delta}^\top \frac{\partial L_\theta(x+\delta,y)}{\partial \delta}}\Bigg|_{\delta=\delta^*}$$

$\square$

### A.2 PROOF OF PROPOSITION 2.2

We provide more detailed and formal statements of Proposition 2.2.

For brevity, consider a $K$-layers fully-connected ReLU network, $f(\theta; x) = f_\theta(x)$ as a function of $\theta$.

$$f(\theta; x) = V^\top D_K W_K D_{K-1} W_{K-1} \cdots D_1 W_K^\top x \tag{14}$$

where the $D_k$ are diagonal matrices dependent on ReLU's activation pattern over the layers, and $W_k$'s and $V$ are the weights (i.e. $\theta$). Note that $f(\theta; x)$ is a piecewise polynomial functions of $\theta$ with finitely many pieces. We further define the directional derivative of a function $g$, along the direction of $\vec{v}$, to be:

$$g'(\theta; \vec{v}) := \lim_{t\downarrow 0} \frac{g(\theta + t\vec{v}) - g(\theta)}{t}.$$

Note that for every direction $\vec{v}$, there exists $\alpha > 0$ such that $f(\theta; x)$ is a polynomial restricted to a line segment $[\theta, \theta + \alpha\vec{v}]$. Thus the above limit exists and the directional derivative is well defined.

We first show the existence of $\vec{v}$ and $t$ for $\mathfrak{l}(\theta_0 + t\vec{v})$ given any $\epsilon$. Let $\mathfrak{l}_{\theta_0,\vec{v},\epsilon}(t) := \sup_{\|\delta\|\leq\epsilon} L(\delta, \theta_0 + t\vec{v})$.

**Proposition A.1.** *For $\epsilon > 0$, $t \in [0, 1]$, and $\theta_0 \in \Theta$, there exists a direction $\vec{v} \in \Theta$, such that the derivative of $\mathfrak{l}_{\theta_0,\vec{v},\epsilon}(t)$ exists and is negative. Moreover, it is given by*

$$\mathfrak{l}'_{\theta_0,\vec{v},\epsilon}(t) = L'(\delta^*, \theta_0; \vec{v}).$$

*Proof.* **[Proof sketch]** Since $\theta_0$ is not a local minimum, there exists a direction $d$, such that $L'(\delta^*, \theta_0; \vec{v}) = \frac{\partial L(\delta^*, \theta_0 + t\vec{v})}{\partial t}$ is negative.

The Danskin theorem provides a way to compute the directional gradient along this direction $\vec{v}$. We basically apply a version of Danskin theorem for directional absolutely continuous maps and semi-continuous maps (Yu, 2012). 1. the constraint set $\{\delta : \|\delta\| \leq \epsilon\}$ is compact; 2. $L(\theta_0 + t\vec{v}; x + \delta, y)$ is piecewise Lipschitz and hence absolutely continuous (an induction argument on the integral representation over the finite pieces). 3. $L(\theta_0 + t\vec{v}; x + \delta, y)$ is continuous on both $\delta$ and along the direction $\vec{v}$ and hence upper semi continuous. Hence we can apply Theorem 1 in Yu (2012). $\square$

Therefore, for any $\epsilon > 0$, if $\theta_0$ is not a local minimum, then there exits a direction $d$, such that for $\theta_1 = \theta_0 + t\vec{v}$ for a proper $t$,

$$\sup_{\|\delta\|\leq\epsilon} L(\delta, \theta_0 + t\vec{v}) < \sup_{\|\delta\|\leq\epsilon} L(\delta, \theta_0). \tag{15}$$

Our next proposition provides an alternative way to increase the margin of $f_\theta$.

**Proposition A.2.** *Assume $f_{\theta_0}$ has a margin $\epsilon_0$, and $\theta_1$ such that $\mathfrak{l}_{\theta_0,\vec{v},\epsilon_0}(t) \leq \mathfrak{l}_{\theta_1,\vec{v},\epsilon_0}(0)$, then $f_{\theta_1}$ has a larger margin than $\epsilon_0$.*

*Proof.* Since $f_{\theta_0}$ has a margin $\epsilon_0$, thus

$$\max_{\|\delta\|\leq\epsilon_0} L(\theta_0; x + \delta, y) = 0$$

Further by $\mathfrak{l}_{\theta_0,\vec{v},\epsilon_0}(t) \leq \mathfrak{l}_{\theta_0,\vec{v},\epsilon_0}(0)$,

$$\sup_{\|\delta\|\leq\epsilon} L(\delta, \theta_0 + t\vec{v}) \leq \sup_{\|\delta\|\leq\epsilon} L(\delta, \theta_0).$$

To see the equality (constraint not binding), we use the following argument. The envelope function's continuity is passed from the continuity of $L(\theta_0; x + \delta, y)$. The inverse image of a closed set under continuous function is closed. If $\delta^*$ lies in the interior of $\max_{\|\delta\|\leq\epsilon_0} L_{\vec{v},\epsilon}(\theta_0; x + \delta, y) \geq 0$, we would have a contradiction. Therefore the constraint is not binding, due to the continuity of the envolope function. By Eq. (15), $\max_{\|\delta\|\leq\epsilon_0} L(\theta_1; x + \delta, y) < 0$. So for the parameter $\theta_1$, $f_{\theta_1}$ has a margin $\epsilon_1 > \epsilon_0$. $\square$

Therefore, the update $\theta_0 \to \theta_1 = \theta_0 + t\vec{v}$ increases the margin of $f_\theta$.

### A.3 PROOF OF PROPOSITION 2.3

*Proof.*

$$L_\theta^{\mathrm{LM}}(x, y) = (\max_{j \neq y} f_\theta^j(x)) - f_\theta^y(x) \tag{16}$$

$$= \log(\exp(\max_{j \neq y} f_\theta^j(x))) - f_\theta^y(x) \tag{17}$$

$$\leq \log(\exp(\sum_{j \neq y} f_\theta^j(x))) - f_\theta^y(x) \tag{18}$$

$$= L_\theta^{\mathrm{SLM}}(x, y) \tag{19}$$

$$\leq \log((K-1)\exp(\max_{j \neq y} f_\theta^j(x))) - f_\theta^y(x) \tag{20}$$

$$= \log(K-1) + (\max_{j \neq y} f_\theta^j(x)) - f_\theta^y(x) \tag{21}$$

$$= \log(K-1) + L_\theta^{\mathrm{LM}}(x, y) \tag{22}$$

Therefore,

$$L_\theta^{\mathrm{SLM}}(x, y) - \log(K-1) \leq L_\theta^{\mathrm{LM}}(x, y) \leq L_\theta^{\mathrm{SLM}}(x, y).$$

$\square$

### A.4 A LEMMA FOR LATER PROOFS

The following lemma helps relate the objective of adversarial training with that of our MMA training. Here, we denote $L_\theta(x + \delta, y)$ as $L(\delta, \theta)$ for brevity.

**Lemma A.1.** *Given $(x, y)$ and $\theta$, assume that $L(\delta, \theta)$ is continuous in $\delta$, then for $\epsilon \geq 0$, and $\rho \geq L(0, \theta) \in Range(L(\delta, \theta))$, it holds that*

$$\min_{L(\delta, \theta) \geq \rho} \|\delta\| = \epsilon \implies \max_{\|\delta\| \leq \epsilon} L(\delta, \theta) = \rho; \tag{23}$$

$$\max_{\|\delta\| \leq \epsilon} L(\delta, \theta) = \rho \implies \min_{L(\delta, \theta) \geq \rho} \|\delta\| \leq \epsilon. \tag{24}$$

*Proof.* **Eq. (23).** We prove this by contradiction. Suppose $\max_{\|\delta\| \leq \epsilon} L(\delta, \theta) > \rho$. When $\epsilon = 0$, this violates our asssumption $\rho \geq L(0, \theta)$ in the theorem. So assume $\epsilon > 0$. Since $L(\delta, \theta)$ is a continuous function defined on a compact set, the maximum is attained by $\bar{\delta}$ such that $\|\bar{\delta}\| \leq \epsilon$ and $L(\bar{\delta}, \theta) > \rho$. Note that $L(\delta, \theta))$ is continuous and $\rho \geq L(0, \theta)$, then there exists $\tilde{\delta} \in \langle 0, \bar{\delta} \rangle$ i.e. the line segment connecting $0$ and $\bar{\delta}$, such that $\|\tilde{\delta}\| < \epsilon$ and $L(\tilde{\delta}, \theta) = \rho$. This follows from the intermediate value theorem by restricting $L(\delta, \theta)$ onto $\langle 0, \bar{\delta} \rangle$. This contradicts $\min_{L(\delta, \theta) \geq \rho} \|\delta\| = \epsilon$.

If $\max_{\|\delta\| \leq \epsilon} L(\delta, \theta) < \rho$, then $\{\delta : \|\delta\| \leq \epsilon\} \subset \{\delta : L(\delta, \theta) < \rho\}$. Every point $p \in \{\delta : \|\delta\| \leq \epsilon\}$ is in the open set $\{\delta : L(\delta, \theta) < \rho\}$, there exists an open ball with some radius $r_p$ centered at $p$ such that $B_{r_p} \subset \{\delta : L(\delta, \theta) < \rho\}$. This forms an open cover for $\{\delta : \|\delta\| \leq \epsilon\}$. Since $\{\delta : \|\delta\| \leq \epsilon\}$ is compact, there is an open finite subcover $\mathcal{U}_\epsilon$ such that: $\{\delta : \|\delta\| \leq \epsilon\} \subset \mathcal{U}_\epsilon \subset \{\delta : L(\delta, \theta) < \rho\}$. Since $\mathcal{U}_\epsilon$ is finite, there exists $h > 0$ such that $\{\delta : \|\delta\| \leq \epsilon + h\} \subset \{\delta : L(\delta, \theta) < \rho\}$. Thus $\{\delta : L(\delta, \theta) \geq \rho\} \subset \{\delta : \|\delta\| > \epsilon + h\}$, contradicting $\min_{L(\delta, \theta) \geq \rho} \|\delta\| = \epsilon$ again.

**Eq. (24).** Assume that $\min_{L(\delta, \theta) \geq \rho} \|\delta\| > \epsilon$, then $\{\delta : L(\delta, \theta) \geq \rho\} \subset \{\delta : \|\delta\| > \epsilon\}$. Taking complementary set of both sides, $\{\delta : \|\delta\| \leq \epsilon\} \subset \{\delta : L(\delta, \theta) < \rho\}$. Therefore, by the compactness of $\{\delta : \|\delta\| \leq \epsilon\}$, $\max_{\|\delta\| \leq \epsilon} L(\delta, \theta) < \rho$, contradiction. $\square$

### A.5 PROOF OF THEOREM 3.1

*Proof.* Recall that $L(\delta, \theta) = L_\theta^{\mathrm{LM}}(x + \delta, y)$, $d_\theta = d_\theta(x, y)$, $\epsilon_\theta^*(\rho) = \min_{\delta:L(\delta,\theta) \geq \rho} \|\delta\|$, and $\rho^* = \max_{\|\delta\| \leq \epsilon} L(\delta, \theta_0) > \max_{\|\delta\| \leq \epsilon} L(\delta, \theta_1)$.

We first prove that $\forall \epsilon$, $\epsilon^*_{\theta_1}(\rho^*) \geq \epsilon^*_{\theta_0}(\rho^*)$ by contradiction. We assume $\epsilon^*_{\theta_1}(\rho^*) < \epsilon^*_{\theta_0}(\rho^*)$. Let $\delta^*_\theta(\rho) = \arg\min_{\delta:L(\delta,\theta)\geq\rho} \|\delta\|$, which is $\|\delta^*_{\theta_1}(\rho^*)\| < \|\delta^*_{\theta_0}(\rho^*)\|$. By Eq. (24), we have $\|\delta^*_{\theta_0}(\rho^*)\| \leq \epsilon$. Therefore, $\|\delta^*_{\theta_1}(\rho^*)\| < \epsilon$. Then there exist a $\delta^\# \in \{\delta : \|\delta\| \leq \epsilon\}$ such that $L(\delta^\#, \theta_1) \geq \rho^*$. This contradicts $\max_{\|\delta\|\leq\epsilon} L(\delta, \theta_1) < \rho^*$. Therefore $\epsilon^*_{\theta_1}(\rho^*) \geq \epsilon^*_{\theta_0}(\rho^*)$.

For 1), $\epsilon = d_{\theta_0}$. By definition of margin in Eq. (1), we have $\rho^* = \max_{\|\delta\|\leq d_{\theta_0}} L(\delta, \theta_0) = 0$. Also by definition of $\epsilon^*_\theta(\rho)$, $\epsilon^*_{\theta_0}(0) = d_{\theta_0}$ and $\epsilon^*_{\theta_1}(0) = d_{\theta_1}$.

For 2), $\epsilon < d_{\theta_0}$. We have $\rho^* = \max_{\|\delta\|\leq\epsilon} L(\delta, \theta_0) \leq \max_{\|\delta\|\leq d_{\theta_0}} L(\delta, \theta_0) = 0$. Therefore $\epsilon^*_{\theta_0}(\rho^*) \leq \epsilon^*_{\theta_0}(0) = d_{\theta_0}$ and $\epsilon^*_{\theta_1}(\rho^*) \leq \epsilon^*_{\theta_1}(0) = d_{\theta_1}$.

For 3), $\epsilon > d_{\theta_0}$. We have $\rho^* = \max_{\|\delta\|\leq\epsilon} L(\delta, \theta_0) \geq \max_{\|\delta\|\leq d_{\theta_0}} L(\delta, \theta_0) = 0$. Therefore $\epsilon^*_{\theta_0}(\rho^*) \geq \epsilon^*_{\theta_0}(0) = d_{\theta_0}$ and $\epsilon^*_{\theta_1}(\rho^*) \geq \epsilon^*_{\theta_1}(0) = d_{\theta_1}$. $\qquad\square$

## B  MORE RELATED WORKS

We next discuss a few related works in details.

**First-order Large Margin:** Previous works (Matyasko & Chau, 2017; Elsayed et al., 2018; Yan et al., 2019) have attempted to use first-order approximation to estimate the input space margin. For first-order methods, the margin will be accurately estimated when the classification function is linear. MMA's margin estimation is exact when the shortest successful perturbation $\delta^*$ can be solved, which is not only satisfied by linear models, but also by a broader range of models, e.g. models that are convex w.r.t. input $x$. This relaxed condition could potentially enable more accurate margin estimation which improves MMA training's performance.

**(Cross-)Lipschitz Regularization:** Sokolic et al. (2017); Tsuzuku et al. (2018) enlarges their margin by controlling the global Lipschitz constant, which in return places a strong constraint on the model and harms its learning capabilities. Instead, our method, alike adversarial training, uses adversarial attacks to estimate the margin to the decision boundary. With a strong method, our estimate is much more precise in the neighborhood around the data point, while being much more flexible due to not relying on a global Lipschitz constraint.

**Hard-Margin SVM (Vapnik, 2013) in the separable case:** Assuming that all the training examples are correctly classified and using our notations on general classifiers, the hard-margin SVM objective can be written as:

$$\max_\theta \left\{ \min_i d_\theta(z_i) \right\} \quad \text{s.t.} \quad L_\theta(z_i) < 0, \forall i. \tag{25}$$

On the other hand, under the same "separable and correct" assumptions, MMA formulation in Eq. (3) can be written as

$$\max_\theta \left\{ \sum_i d_\theta(z_i) \right\} \quad \text{s.t.} \quad L_\theta(z_i) < 0, \forall i, \tag{26}$$

which is maximizing the *average* margin rather than the *minimum* margin in SVM. Note that the theorem on gradient calculation of the margin in Section 2.1 also applies to the SVM formulation of differentiable functions. Because of this, we can also use SGD to solve the following "SVM-style" formulation:

$$\max_\theta \left\{ \min_{i\in\mathcal{S}^+_\theta} d_\theta(z_i) - \sum_{j\in\mathcal{S}^-_\theta} J_\theta(z_j) \right\}. \tag{27}$$

As our focus is using MMA to improve adversarial robustness which involves maximizing the average margin, we delay the maximization of minimum margin to future work.

**Maximizing Robust Radius for Randomized Smoothing:** Randomized smoothing (Cohen et al., 2019) is an effective technique for certifying the $\ell_2$ robust radius of a "smoothed classifier" with high probability bound. Here the certified robust radius is a lower bound of the input space margin of the "smoothed classifier". Zhai et al. (2020) introduce a method to directly maximizing this certified robust radius.

**Bayesian Adversarial Learning:** Ye & Zhu (2018) introduce a full Bayesian treatment of adversarial training. In practice, the model is trained to be robust under adversarial perturbations with

different magnitudes. Although both their and our methods adversarially train the model with different perturbation magnitudes, the motivations are completely different.

**Dynamic Adversarial Training:** Wang et al. (2019) improve adversarial training's convergence by dynamically adjusting the strength (the number of iterations) of PGD attacks, based on its proposed First-Order Stationary Condition. In a sense, MMA training is also dynamically adjusting the attack strength. However, our aim is margin maximization and the strength here represents the magnitude of the perturbation.

### B.1 DETAILED COMPARISON WITH ADVERSARIAL TRAINING WITH DDN

For $\ell_2$ robustness, we also compare to models adversarially trained on the "Decoupled Direction and Norm" (DDN) attack (Rony et al., 2018), which is concurrent to our work. The DDN attack aims to achieve successful perturbation with minimal $\ell_2$ norm, thus, DDN could be used as a drop-in replacement for the AN-PGD attack for MMA training. We performed evaluations on the downloaded [8] DDN trained models.

The DDN MNIST model is a larger ConvNet with similar structure to our LeNet5, and the CIFAR10 model is wideresnet-28-10, which is similar but larger than the wideresnet-28-4 that we use.

DDN training, "training on adversarial examples generated by the DDN attack", differs from MMA in the following ways. When the DDN attack does not find a successful adversarial example, it returns the clean image, and the model will use it for training. In MMA, when a successful adversarial example cannot be found, it is treated as a perturbation with very large magnitude, which will be ignored by the hinge loss when we calculate the gradient for this example. Also, in DDN training, there exists a maximum norm of the perturbation. This maximum norm constraint does not exist for MMA training. When a perturbation is larger than the hinge threshold, it will be ignored by the hinge loss. There also are differences in training hyperparameters, which we refer the reader to Rony et al. (2018) for details.

Despite these differences, in our experiments MMA training achieves similar performances under the $\ell_2$ cases. While DDN attack and training only focus on $\ell_2$ cases, we also show that the MMA training framework provides significant improvements over PGD training in the $\ell_\infty$ case.

## C DETAILED SETTINGS FOR TRAINING

With respect to the weights on $L^{\mathrm{CE}}$ and $L^{\mathrm{MMA}}$ in Eq. (8), we tested 3 pairs of weights, (1/3, 2/3), (1/2, 1/2) and (2/3, 1/3), in our initial CIFAR10 $\ell_\infty$ experiments. We observed that (1/3, 2/3), namely (1/3 for $L^{\mathrm{CE}}$ and 2/3 for $L^{\mathrm{MMA}}$) gives better performance. We then fixed it and use the same values for all the other experiments in the paper, including the MNIST experiments and $\ell_2$ experiments.

We train LeNet5 models for the MNIST experiments and use wide residual networks (Zagoruyko & Komodakis, 2016) with depth 28 and widen factor 4 for all the CIFAR10 experiments. For all the experiments, we monitor the average margin from AN-PGD on the validation set and choose the model with largest average margin from the sequence of checkpoints during training. The validation set contains first 5000 images of training set. It is only used to monitor training progress and not used in training. Here all the models are trained and tested under the same type of norm constraints, namely if trained on $\ell_\infty$, then tested on $\ell_\infty$; if trained on $\ell_2$, then tested on $\ell_2$.

The LeNet5 is composed of 32-channel conv filter + ReLU + size 2 max pooling + 64-channel conv filter + ReLU + size 2 max pooling + fc layer with 1024 units + ReLU + fc layer with 10 output classes. We do not preprocess MNIST images before feeding into the model.

For training LeNet5 on all MNIST experiments, for both PGD and MMA training, we use the Adam optimizer with an initial learning rate of 0.0001 and train for 100000 steps with batch size 50. In our initial experiments, we tested different initial learning rate at 0.0001, 0.001, 0.01, and 0.1 and do not find noticeable differences.

---

[8]`github.com/jeromerony/fast_adversarial`

We use the WideResNet-28-4 as described in Zagoruyko & Komodakis (2016) for our experiments, where 28 is the depth and 4 is the widen factor. We use "per image standardization" [9] to preprocess CIFAR10 images, following Madry et al. (2017).

For training WideResNet on CIFAR10 variants, we use stochastic gradient descent with momentum 0.9 and weight decay 0.0002. We train 50000 steps in total with batch size 128. The learning rate is set to 0.3 at step 0, 0.09 at step 20000, 0.03 at step 30000, and 0.009 at step 40000. This setting is the same for PGD and MMA training. In our initial experiments, we tested different learning rate at 0.03, 0.1, 0.3, and 0.6, and kept using 0.3 for all our later experiments. We also tested a longer training schedule, following Madry et al. (2017), where we train 80000 steps with different learning rate schedules. We did not observe improvement with this longer training, therefore kept using the 50000 steps training.

For models trained on MNIST, we use 40-step PGD attack with the soft logit margin (SLM) loss defined in Section 3, for CIFAR10 we use 10 step-PGD, also with the SLM loss. For both MNIST and CIFAR10, the step size of PGD attack at training time is $\frac{2.5\epsilon}{\text{number of steps}}$. In AN-PGD, we always perform 10 step bisection search after PGD, with the SLM loss. For AN-PGD, the maximum perturbation length is always 1.05 times the hinge threshold: $\epsilon_{max} = 1.05 d_{\max}$. The initial perturbation length at the first epoch, $\epsilon_{init}$, have different values under different settings. $\epsilon_{init} = 0.5$ for MNIST $\ell_2$, $\epsilon_{init} = 0.1$ for MNIST $\ell_\infty$, $\epsilon_{init} = 0.5$ for CIFAR10 $\ell_2$, $\epsilon_{init} = 0.05$ for CIFAR10 $\ell_2$. In epochs after the first, $\epsilon_{init}$ will be set to the margin of the same example from last epoch.

**Trained models**: Various PGD/PGDLS models are trained with different perturbation magnitude $\epsilon$, denoted by PGD-$\epsilon$ or PGDLS-$\epsilon$. PGD-ens/PGDLS-ens represents the ensemble of PGD/PGDLS trained models with different $\epsilon$'s. The ensemble makes prediction by majority voting on label predictions, and uses softmax scores as the tie breaker. We perform MMA training with different hinge thresholds $d_{\max}$, also with/without the additional clean loss (see next section for details). We use OMMA to represent training with only $L_\theta^{\text{MMA}}$ in Eq. (7), and MMA to represent training with the combined loss in Eq. (8). When train For each $d_{\max}$ value, we train two models with different random seeds, which serves two purposes: 1) confirming the performance of MMA trained models are not significantly affected by random initialization; 2) to provide transfer attacks from an "identical" model. As such, MMA trained models are named as OMMA/MMA-$d_{\max}$-seed. Models shown in the main body correspond to those with seed "sd0".

For MNIST-$\ell_\infty$, we train PGD/PGDLS models with $\epsilon = 0.1, 0.2, 0.3, 0.4, 0.45$. We also train OMMA/MMA models with $d_{\max} = 0.45$, each with two random seeds. We also consider a standard model trained on clean images, PGD-ens, PGDLS-ens, and the PGD trained model from Madry et al. (2017). Therefore, we have in total $m = 2 \times 5 + 2 \times 2 + 4 = 18$ models under MNIST-$\ell_\infty$.

For MNIST-$\ell_2$, we train PGD/PGDLS models with $\epsilon = 1.0, 2.0, 3.0, 4.0$. We also train OMMA/MMA models with $d_{\max} = 2.0, 4.0, 6.0$, each with two random seeds. We also consider a standard model trained on clean images, PGD-ens, PGDLS-ens, and the DDN trained model from Rony et al. (2018). Therefore, we have in total $m = 2 \times 4 + 2 \times 3 \times 2 + 4 = 24$ models under MNIST-$\ell_2$.

For CIFAR10-$\ell_\infty$, we train PGD/PGDLS models with $\epsilon = 4, 8, 12, 16, 20, 24, 28, 32$. We also train OMMA/MMA models with $d_{\max} = 12, 20, 32$, each with two random seeds. We also consider a standard model trained on clean images, PGD-ens, PGDLS-ens, and the PGD trained model from Madry et al. (2017). Therefore, we have in total $m = 2 \times 8 + 2 \times 3 \times 2 + 4 = 32$ models under CIFAR10-$\ell_\infty$.

For CIFAR10-$\ell_2$, we train PGD/PGDLS models with $\epsilon = 0.5, 1.0, 1.5, 2.0, 2.5$. We also train OMMA/MMA models with $d_{\max} = 1.0, 2.0, 3.0$, each with two random seeds. We also consider a standard model trained on clean images, PGD-ens, PGDLS-ens, and the DDN trained model from Rony et al. (2018). Therefore, we have in total $m = 2 \times 5 + 2 \times 3 \times 2 + 4 = 26$ models under MNIST-$\ell_2$.

With regard to ensemble models, for MNIST-$\ell_2$ PGD/PGDLS-ens, CIFAR10-$\ell_2$ PGD/PGDLS-ens, MNIST-$\ell_\infty$PGDLS-ens, and CIFAR10-$\ell_\infty$ PGDLS-ens, they all use the PGD (or PGDLS) models

---

[9]Description can be found at https://www.tensorflow.org/api_docs/python/tf/image/per_image_standardization. We implemented our own version in PyTorch.

trained at all testing (attacking) $\epsilon$'s. For CIFAR10-$\ell_\infty$ PGD-ens, PGD-24,28,32 are excluded for the same reason.

## D  DETAILED SETTINGS OF ATTACKS

For both $\ell_\infty$ and $\ell_2$ PGD attacks, we use the implementation from the AdverTorch toolbox (Ding et al., 2019b). Regarding the loss function of PGD, we use both the cross entropy (CE) loss and the Carlini & Wagner (CW) loss. [10]

As previously stated, each model will have $N$ whitebox PGD attacks on them, $N/2$ of them are CE-PGD attacks, and the other $N/2$ are CW-PGD attacks. Recall that $N = 50$ for MNIST and $N = 10$ for CIFAR10. At test time, all the PGD attack run 100 iterations. We manually tune the step size parameter on a few MMA and PGD models and then fix them thereafter. The step size for MNIST-$\ell_\infty$ when $\epsilon = 0.3$ is 0.0075, the step size for CIFAR10-$\ell_\infty$ when $\epsilon = 8/255$ is $2/255$, the step size for MNIST-$\ell_2$ when $\epsilon = 1.0$ is 0.25, the step size for CIFAR10-$\ell_2$ when $\epsilon = 1.0$ is 0.25. For other $\epsilon$ values, the step size is linearly scaled accordingly.

The ensemble model we considered uses the majority vote for prediction, and uses softmax score as the tie breaker. So it is not obvious how to perform CW-PGD and CE-PGD directly on them. Here we take 2 strategies. The first one is a naive strategy, where we minimize the sum of losses of all the models used in the ensemble. Here, similar to attacking single models, we CW and CE loss here and perform the same number attacks.

The second strategy is still a PGD attack with a customized loss towards attacking ensemble models. For the group of classifiers in the ensemble, at each PGD step, if less than half of the classifiers give wrong classification, we sum up the CW losses from correct classifiers as the loss for the PGD attack. If more than half of the classifiers give wrong classification, then we find the wrong prediction that appeared most frequently among classifiers, and denote it as label0, with its corresponding logit, logit0. For each classifier, we then find the largest logit that is not logit0, denoted as logit1. The loss we maximize, in the PGD attack, is the sum of "logit1 - logit0" from each classifier. Using this strategy, we perform additional (compared to attacking single models) whitebox PGD attacks on ensemble models. For MNIST, we perform 50 repeated attacks, for CIFAR10 we perform 10. These are also 100-step PGD attacks.

We expect more carefully designed attacks could work better on ensembles, but we delay it to future work.

For the SPSA attack (Uesato et al., 2018), we run the attack for 100 iterations with perturbation size 0.01 (for gradient estimation), Adam learning rate 0.01, stopping threshold -5.0 and 2048 samples for each gradient estimate. For CIFAR10-$\ell_\infty$, we use $\epsilon = 8/255$. For MNIST-$\ell_\infty$, we use $\epsilon = 0.3$.

## E  EFFECTS OF ADDING CLEAN LOSS IN ADDITION TO THE MMA LOSS

We further examine the effectiveness of adding a clean loss term to the MMA loss. We represent MMA trained models with the MMA loss in Eq. (7) as *MMA-$d_{\max}$*. In Section 2.4, we introduced MMAC-$d_{\max}$ models to resolve MMA-$d_{\max}$ model's problem of having flat input space loss landscape and showed its effectiveness qualitatively. Here we demonstrate the quantitative benefit of adding the clean loss.

We observe that models trained with the MMA loss in Eq. (7) have certain degrees of TransferGaps. The term *TransferGaps* represents the difference between robust accuracy under "combined (whitebox+transfer) attacks" and under "only whitebox PGD attacks". In other words, it is the additional attack success rate that transfer attacks bring. For example, OMMA-32 achieves 53.70% under whitebox PGD attacks, but achieves a lower robust accuracy at 46.31% under combined (whitebox+transfer) attacks, therefore it has a TransferGap of 7.39% (See Appendix F for full results.). After adding the clean loss, MMA-32 reduces its TransferGap at $\epsilon = 8/255$ to 3.02%. This corresponds to our observation in Section 2.4 that adding clean loss makes the loss landscape more tilted, such that whitebox PGD attacks can succeed more easily.

---

[10]The CW loss is almost equivalent to the logit margin (LM) loss. We use CW loss for the ease of comparing with literature. Here $CW(x) = \min\{\max_{j \neq y} f_j(x) - f_y(x), 0\}$. When the classification is correct, CW and LM loss have the same gradient.

Recall that MMA trained models are robust to gradient free attacks, as described in Section 4.3. Therefore, robustness of MMA trained models and the TransferGaps are likely not due to gradient masking.

We also note that TransferGaps for both MNIST-$\ell_\infty$ and $\ell_2$ cases are almost zero for the MMA trained models, indicating that TransferGaps, observed on CIFAR10 cases, are not solely due to the MMA algorithm, data distributions (MNIST vs CIFAR10) also play an important role.

Another interesting observation is that, for MMA trained models trained on CIFAR10, adding additional clean loss results in a decrease in clean accuracy and an increase in the average robust accuracy, e.g. OMMA-32 has ClnAcc 86.11%, and AvgRobAcc 28.36%, whereas MMA-32 has ClnAcc 84.36%, and AvgRobAcc 29.39%. The fact that "adding additional clean loss results in a model with lower accuracy and more robustness" seems counter-intuitive. However, it actually confirms our motivation and reasoning of the additional clean loss: it makes the input space loss landscape steeper, which leads to stronger adversaries at training time, which in turn poses more emphasis on "robustness training", instead of clean accuracy training.

# F    FULL RESULTS AND TABLES

We present all the empirical results in Table 4 to 15. Specifically, we show model performances under combined (whitebox+transfer) attacks in Tables 4 to 7. This is our proxy for true robustness measure. We show model performances under only whitebox PGD attacks in Tables 8 to 11. We show TransferGaps in Tables 12 to 15.

In these tables, PGD-Madry et al. models are the "secret" models downloaded from `https://github.com/MadryLab/mnist_challenge` and `https://github.com/MadryLab/cifar10_challenge/`. DDN-Rony et al. models are downloaded from `https://github.com/jeromerony/fast_adversarial/`. TRADES models are downloaded from `https://github.com/yaodongyu/TRADES`.

For MNIST PGD-Madry et al. models, our whitebox attacks brings the robust accuracy at $\epsilon = 0.3$ down to 89.79%, which is at the same level with the reported 89.62% on the website, also with 50 repeated random initialized PGD attacks. For CIFAR10 PGD-Madry et al. models, our whitebox attacks brings the robust accuracy at $\epsilon = 8/255$ down to 44.70%, which is stronger than the reported 45.21% on the website, with 10 repeated random initialized 20-step PGD attacks. As our PGD attacks are 100-step, this is not surprising.

We also compared TRADES with MMA trained models under $\ell_\infty$ attacks. On MNIST, overall TRADES outperforms MMA except that TRADES fails completely at large perturbation length of 0.4. This may be because TRADES is trained with attacking length $\epsilon = 0.3$, and therefore cannot defend larger attacks. On CIFAR10, we compare MMA-32 with TRADES as their clean accuracies are similar. Similar to the results on MNIST, we see that TRADES outperforms MMA on the attacks of lengths that are less than $\epsilon = 12/255$, but it sacrifices the robustness under larger attacks. We also note that MMA training and TRADES' idea of optimizing a calibration loss are progresses in orthogonal directions and could potentially be combined.

As we mentioned previously, DDN training can be seen as a specific instantiation of the general MMA training idea, and the DDN-Rony et al. models indeed performs very similar to MMA trained models when $d_{\max}$ is set relatively low. Therefore, we do not discuss the performance of DDN-Rony et al. separately.

In Section 4, we have mainly discussed different phenomena under the case of CIFAR10-$\ell_\infty$. For CIFAR10-$\ell_2$, we see very similar patterns in Tables 7, 11 and 15. These include

- MMA training is fairly stable to $d_{\max}$, and achieves good robustness-accuracy trade-offs. On the other hand, to achieve good AvgRobAcc, PGD/PGDLS trained models need to have large sacrifices on clean accuracies.
- Adding additional clean loss increases the robustness of the model, reduce TransferGap, at a cost of slightly reducing clean accuracy.

As a simpler datasets, different adversarial training algorithms, including MMA training, have very different behaviors on MNIST as compared to CIFAR10.

We first look at MNIST-$\ell_\infty$. Similar to CIFAR10 cases, PGD training is incompetent on large $\epsilon$'s, e.g. PGD-0.4 has significant drop on clean accuracy (to 96.64%) and PGD-0.45 fails to train. PGDLS training, on the other hand, is able to handle large $\epsilon$'s training very well on MNIST-$\ell_\infty$, and MMA training does not bring extra benefit on top of PGDLS. We suspect that this is due to the "easiness" of this specific task on MNIST, where finding proper $\epsilon$ for each individual example is not necessary, and a global scheduling of $\epsilon$ is enough. We note that this phenomenon confirms our understanding of adversarial training from the margin maximization perspective in Section 3.

Under the case of MNIST-$\ell_2$, we notice that MMA training almost does not need to sacrifice clean accuracy in order to get higher robustness. All the models with $d_{\max} \geq 4.0$ behaves similarly w.r.t. both clean and robust accuracies. Achieving 40% robust accuracy at $\epsilon = 3.0$ seems to be the robustness limit of MMA trained models. On the other hand, PGD/PGDLS models are able to get higher robustness at $\epsilon = 3.0$ with robust accuracy of 44.5%, although with some sacrifices to clean accuracy. This is similar to what we have observed in the case of CIFAR10.

We notice that on both MNIST-$\ell_\infty$ and MNIST-$\ell_2$, unlike CIFAR10 cases, PGD(LS)-ens model performs poorly in terms of robustness. This is likely due to that PGD trained models on MNIST usually have a very sharp robustness drop when the $\epsilon$ used for attacking is larger than the $\epsilon$ used for training.

Another significant differences between MNIST cases and CIFAR10 cases is that TransferGaps are very small for OMMA/MMA trained models on MNIST cases. This again is likely due to that MNIST is an "easier" dataset. It also indicates that the TransferGap is not purely due to the MMA training algorithm, it is also largely affected by the property of datasets. Although previous literature (Ding et al., 2019a; Zhang et al., 2019c) also discusses related topics on the difference between MNIST and CIFAR10 w.r.t. adversarial robustness, they do not directly explain the observed phenomena here. We delay a thorough understanding of this topic to future work.

### F.1 ADDITIONAL EVALUATION WITH THE CW-$\ell_2$ ATTACK

We run the CW-$\ell_2$ attack (Carlini & Wagner, 2017) on the first 1000 test examples for models trained with $\ell_2$ attacks. This gives the minimum $\ell_2$ norm a perturbation needs to change the prediction. For each model, we show the clean accuracy, also mean, median, 25th percentile and 75th percentile of these minimum distances in Tables 2 and 3 below.

Looking at CW-$\ell_2$ results, we have very similar observations as compared to the observations we made based on robust accuracies (under PGD attacks) at different $\epsilon$'s and the average robust accuracy.

On MNIST, for MMA trained models as $d_{\max}$ increases from 2.0 to 6.0, the mean minimum distance goes from 2.15 to 2.76, with the clean accuracy only drops to 97.7% from 99.3%. On the contrary, when we perform PGD/PGDLS with a larger $\epsilon$, e.g. $\epsilon = 4.0$, the clean accuracy drops significantly to 91.8%, with the mean minimum distance increased to 2.53. Therefore MMA training achieves better performance on both clean accuracy and robustness under the MNIST-$\ell_2$ case.

On CIFAR10, we observed that MMA training is fairly stable to $d_{\max}$, and achieves good robustness-accuracy trade-offs. With carefully chosen $\epsilon$ value, PGD-1.0/PGDLS-1.0 models can also achieve similar robustness-accuracy tradeoffs as compared to MMA training. However, when we perform PGD training with a larger $\epsilon$, the clean accuracy drops significantly.

Table 2: CW-$\ell_2$ attack results on models trained on MNIST with $\ell_2$-norm constrained attacks.

| MNIST Model | Cln Acc | Avg Norm | 25% Norm | Median Norm | 75% Norm |
|---|---|---|---|---|---|
| STD | 99.0 | 1.57 | 1.19 | 1.57 | 1.95 |
| PGD-1.0 | 99.2 | 1.95 | 1.58 | 1.95 | 2.33 |
| PGD-2.0 | 98.6 | 2.32 | 1.85 | 2.43 | 2.82 |
| PGD-3.0 | 96.8 | 2.54 | 1.79 | 2.76 | 3.39 |
| PGD-4.0 | 91.8 | 2.53 | 1.38 | 2.75 | 3.75 |
| PGDLS-1.0 | 99.3 | 1.88 | 1.53 | 1.88 | 2.24 |
| PGDLS-2.0 | 99.3 | 2.23 | 1.82 | 2.30 | 2.68 |
| PGDLS-3.0 | 96.9 | 2.51 | 1.81 | 2.74 | 3.29 |
| PGDLS-4.0 | 91.8 | 2.53 | 1.46 | 2.79 | 3.72 |
| MMA-2.0 | 99.3 | 2.15 | 1.81 | 2.22 | 2.58 |
| MMA-4.0 | 98.5 | 2.67 | 1.86 | 2.72 | 3.52 |
| MMA-6.0 | 97.7 | 2.76 | 1.84 | 2.72 | 3.61 |

Table 3: CW-$\ell_2$ attack results on models trained on CIFAR10 with $\ell_2$-norm constrained attacks.

| CIFAR10 Model | Cln Acc | Avg Norm | 25% Norm | Median Norm | 75% Norm |
|---|---|---|---|---|---|
| STD | 95.1 | 0.09 | 0.05 | 0.08 | 0.13 |
| PGD-0.5 | 89.2 | 0.80 | 0.32 | 0.76 | 1.21 |
| PGD-1.0 | 83.8 | 1.01 | 0.30 | 0.91 | 1.62 |
| PGD-1.5 | 76.0 | 1.11 | 0.04 | 0.96 | 1.87 |
| PGD-2.0 | 71.6 | 1.16 | 0.00 | 0.96 | 2.03 |
| PGD-2.5 | 65.2 | 1.19 | 0.00 | 0.88 | 2.10 |
| PGDLS-0.5 | 90.5 | 0.79 | 0.34 | 0.76 | 1.17 |
| PGDLS-1.0 | 83.8 | 1.00 | 0.30 | 0.91 | 1.59 |
| PGDLS-1.5 | 77.6 | 1.10 | 0.11 | 0.97 | 1.83 |
| PGDLS-2.0 | 73.1 | 1.16 | 0.00 | 0.98 | 2.02 |
| PGDLS-2.5 | 66.0 | 1.19 | 0.00 | 0.88 | 2.13 |
| MMA-1.0 | 88.4 | 0.85 | 0.32 | 0.80 | 1.27 |
| MMA-2.0 | 84.2 | 1.03 | 0.26 | 0.92 | 1.62 |
| MMA-3.0 | 81.2 | 1.09 | 0.21 | 0.92 | 1.73 |

Table 4: Accuracies of models trained on MNIST with $\ell_\infty$-norm constrained attacks. These robust accuracies are calculated under both combined (whitebox+transfer) PGD attacks. sd0 and sd1 indicate 2 different random seeds.

| MNIST Model | Cln Acc | AvgAcc | AvgRobAcc | RobAcc under different $\epsilon$, combined (whitebox+transfer) attacks | | | |
| --- | --- | --- | --- | --- | --- | --- | --- |
| | | | | 0.1 | 0.2 | 0.3 | 0.4 |
| STD | 99.21 | 35.02 | 18.97 | 73.58 | 2.31 | 0.00 | 0.00 |
| PGD-0.1 | 99.40 | 48.85 | 36.22 | 96.35 | 48.51 | 0.01 | 0.00 |
| PGD-0.2 | 99.22 | 57.92 | 47.60 | 97.44 | 92.12 | 0.84 | 0.00 |
| PGD-0.3 | 98.96 | 76.97 | 71.47 | 97.90 | 96.00 | 91.76 | 0.22 |
| PGD-0.4 | 96.64 | 89.37 | 87.55 | 94.69 | 91.57 | 86.49 | 77.47 |
| PGD-0.45 | 11.35 | 11.35 | 11.35 | 11.35 | 11.35 | 11.35 | 11.35 |
| PGDLS-0.1 | 99.43 | 46.85 | 33.71 | 95.41 | 39.42 | 0.00 | 0.00 |
| PGDLS-0.2 | 99.38 | 58.36 | 48.10 | 97.38 | 89.49 | 5.53 | 0.00 |
| PGDLS-0.3 | 99.10 | 76.56 | 70.93 | 97.97 | 95.66 | 90.09 | 0.00 |
| PGDLS-0.4 | 98.98 | 93.07 | 91.59 | 98.12 | 96.29 | 93.01 | 78.96 |
| PGDLS-0.45 | 98.89 | 94.74 | 93.70 | 97.91 | 96.34 | 93.29 | 87.28 |
| MMA-0.45-sd0 | 98.95 | 94.13 | 92.93 | 97.87 | 96.01 | 92.59 | 85.24 |
| MMA-0.45-sd1 | 98.90 | 94.04 | 92.82 | 97.82 | 96.00 | 92.63 | 84.83 |
| OMMA-0.45-sd0 | 98.98 | 93.94 | 92.68 | 97.90 | 96.05 | 92.35 | 84.41 |
| OMMA-0.45-sd1 | 99.02 | 94.03 | 92.78 | 97.93 | 96.02 | 92.44 | 84.73 |
| PGD-ens | 99.28 | 57.98 | 47.65 | 97.25 | 89.99 | 3.37 | 0.00 |
| PGDLS-ens | 99.34 | 59.04 | 48.96 | 97.48 | 90.40 | 7.96 | 0.00 |
| PGD-Madry et al. | 98.53 | 76.04 | 70.41 | 97.08 | 94.83 | 89.64 | 0.11 |
| TRADES | 99.47 | 77.94 | 72.56 | 98.81 | 97.27 | 94.11 | 0.05 |

Table 5: Accuracies of models trained on CIFAR10 with $\ell_\infty$-norm constrained attacks. These robust accuracies are calculated under both combined (whitebox+transfer) PGD attacks. sd0 and sd1 indicate 2 different random seeds.

| CIFAR10 Model | Cln Acc | AvgAcc | AvgRobAcc | RobAcc under different $\epsilon$, combined (whitebox+transfer) attacks | | | | | | | |
| --- | --- | --- | --- | --- | --- | --- | --- | --- | --- | --- | --- |
| | | | | 4 | 8 | 12 | 16 | 20 | 24 | 28 | 32 |
| STD | 94.92 | 10.55 | 0.00 | 0.00 | 0.00 | 0.00 | 0.00 | 0.00 | 0.00 | 0.00 | 0.00 |
| PGD-4 | 90.44 | 22.95 | 14.51 | 66.31 | 33.49 | 12.22 | 3.01 | 0.75 | 0.24 | 0.06 | 0.01 |
| PGD-8 | 85.14 | 27.27 | 20.03 | 67.73 | 46.47 | 26.63 | 12.33 | 4.69 | 1.56 | 0.62 | 0.22 |
| PGD-12 | 77.86 | 28.51 | 22.34 | 63.88 | 48.22 | 32.13 | 18.67 | 9.48 | 4.05 | 1.56 | 0.70 |
| PGD-16 | 68.86 | 28.28 | 23.21 | 57.99 | 46.09 | 33.64 | 22.73 | 13.37 | 7.01 | 3.32 | 1.54 |
| PGD-20 | 61.06 | 27.34 | 23.12 | 51.72 | 43.13 | 33.73 | 24.55 | 15.66 | 9.05 | 4.74 | 2.42 |
| PGD-24 | 10.90 | 9.95 | 9.83 | 10.60 | 10.34 | 10.11 | 10.00 | 9.89 | 9.69 | 9.34 | 8.68 |
| PGD-28 | 10.00 | 10.00 | 10.00 | 10.00 | 10.00 | 10.00 | 10.00 | 10.00 | 10.00 | 10.00 | 10.00 |
| PGD-32 | 10.00 | 10.00 | 10.00 | 10.00 | 10.00 | 10.00 | 10.00 | 10.00 | 10.00 | 10.00 | 10.00 |
| PGDLS-4 | 89.87 | 22.39 | 13.96 | 63.98 | 31.92 | 11.47 | 3.32 | 0.68 | 0.16 | 0.08 | 0.05 |
| PGDLS-8 | 85.63 | 27.20 | 19.90 | 67.96 | 46.19 | 26.19 | 12.22 | 4.51 | 1.48 | 0.44 | 0.21 |
| PGDLS-12 | 79.39 | 28.45 | 22.08 | 64.62 | 48.08 | 31.34 | 17.86 | 8.69 | 3.95 | 1.48 | 0.65 |
| PGDLS-16 | 70.68 | 28.44 | 23.16 | 59.43 | 47.00 | 33.64 | 21.72 | 12.66 | 6.54 | 2.98 | 1.31 |
| PGDLS-20 | 65.81 | 27.60 | 22.83 | 54.96 | 44.39 | 33.13 | 22.53 | 13.80 | 7.79 | 4.08 | 1.95 |
| PGDLS-24 | 58.36 | 26.53 | 22.55 | 49.05 | 41.13 | 32.10 | 23.76 | 15.70 | 9.66 | 5.86 | 3.11 |
| PGDLS-28 | 50.07 | 24.20 | 20.97 | 40.71 | 34.61 | 29.00 | 22.77 | 16.83 | 11.49 | 7.62 | 4.73 |
| PGDLS-32 | 38.80 | 19.88 | 17.52 | 26.16 | 24.96 | 23.22 | 19.96 | 16.22 | 12.92 | 9.82 | 6.88 |
| MMA-12-sd0 | 88.59 | 26.87 | 19.15 | 67.96 | 43.42 | 24.07 | 11.45 | 4.27 | 1.43 | 0.45 | 0.16 |
| MMA-12-sd1 | 88.91 | 26.23 | 18.39 | 67.08 | 42.97 | 22.57 | 9.76 | 3.37 | 0.92 | 0.35 | 0.12 |
| MMA-20-sd0 | 86.56 | 28.86 | 21.65 | 66.92 | 46.89 | 29.83 | 16.55 | 8.14 | 3.25 | 1.17 | 0.43 |
| MMA-20-sd1 | 85.87 | 28.72 | 21.57 | 65.44 | 46.11 | 29.96 | 17.30 | 8.27 | 3.60 | 1.33 | 0.56 |
| MMA-32-sd0 | 84.36 | 29.39 | 22.51 | 64.82 | 47.18 | 31.49 | 18.91 | 10.16 | 4.77 | 1.97 | 0.81 |
| MMA-32-sd1 | 84.76 | 29.08 | 22.11 | 64.41 | 45.95 | 30.36 | 18.24 | 9.85 | 4.99 | 2.20 | 0.92 |
| OMMA-12-sd0 | 88.52 | 26.31 | 18.54 | 66.96 | 42.58 | 23.22 | 10.29 | 3.43 | 1.24 | 0.46 | 0.13 |
| OMMA-12-sd1 | 87.82 | 26.24 | 18.54 | 66.23 | 43.10 | 23.57 | 10.32 | 3.56 | 1.04 | 0.38 | 0.14 |
| OMMA-20-sd0 | 87.06 | 27.41 | 19.95 | 66.54 | 45.39 | 26.29 | 13.09 | 5.32 | 1.96 | 0.79 | 0.23 |
| OMMA-20-sd1 | 87.44 | 27.77 | 20.31 | 66.28 | 45.60 | 27.33 | 14.00 | 6.04 | 2.23 | 0.74 | 0.25 |
| OMMA-32-sd0 | 86.11 | 28.36 | 21.14 | 66.02 | 46.31 | 28.88 | 15.98 | 7.44 | 2.94 | 1.12 | 0.45 |
| OMMA-32-sd1 | 86.36 | 28.75 | 21.55 | 66.86 | 47.12 | 29.63 | 16.09 | 7.56 | 3.38 | 1.31 | 0.47 |
| PGD-ens | 87.38 | 28.10 | 20.69 | 64.59 | 46.95 | 28.88 | 15.10 | 6.35 | 2.35 | 0.91 | 0.39 |
| PGDLS-ens | 76.73 | 29.52 | 23.62 | 60.52 | 48.21 | 35.06 | 22.14 | 12.28 | 6.17 | 3.14 | 1.43 |
| PGD-Madry et al. | 87.14 | 27.22 | 19.73 | 68.01 | 44.68 | 25.03 | 12.15 | 5.18 | 1.95 | 0.64 | 0.23 |
| TRADES | 84.92 | 30.46 | 23.65 | 70.96 | 52.92 | 33.04 | 18.23 | 8.34 | 3.57 | 1.4 | 0.69 |

Table 6: Accuracies of models trained on MNIST with $\ell_2$-norm constrained attacks. These robust accuracies are calculated under both combined (whitebox+transfer) PGD attacks. sd0 and sd1 indicate 2 different random seeds.

| MNIST Model | Cln Acc | AvgAcc | AvgRobAcc | RobAcc under different $\epsilon$, combined (whitebox+transfer) attacks | | | |
| --- | --- | --- | --- | --- | --- | --- | --- |
| | | | | 1.0 | 2.0 | 3.0 | 4.0 |
| STD | 99.21 | 41.84 | 27.49 | 86.61 | 22.78 | 0.59 | 0.00 |
| PGD-1.0 | 99.30 | 48.78 | 36.15 | 95.06 | 46.84 | 2.71 | 0.00 |
| PGD-2.0 | 98.76 | 56.14 | 45.48 | 94.82 | 72.70 | 14.20 | 0.21 |
| PGD-3.0 | 97.14 | 60.36 | 51.17 | 90.01 | 71.03 | 38.93 | 4.71 |
| PGD-4.0 | 93.41 | 59.52 | 51.05 | 82.34 | 66.25 | 43.44 | 12.18 |
| PGDLS-1.0 | 99.39 | 47.61 | 34.66 | 94.33 | 42.44 | 1.89 | 0.00 |
| PGDLS-2.0 | 99.09 | 54.73 | 43.64 | 95.22 | 69.33 | 10.01 | 0.01 |
| PGDLS-3.0 | 97.52 | 60.13 | 50.78 | 90.86 | 71.91 | 36.80 | 3.56 |
| PGDLS-4.0 | 93.68 | 59.49 | 50.95 | 82.67 | 67.21 | 43.68 | 10.23 |
| MMA-2.0-sd0 | 99.27 | 53.85 | 42.50 | 95.59 | 68.37 | 6.03 | 0.01 |
| MMA-2.0-sd1 | 99.28 | 54.34 | 43.10 | 95.78 | 68.18 | 8.45 | 0.00 |
| MMA-4.0-sd0 | 98.71 | 62.25 | 53.13 | 93.93 | 74.01 | 39.34 | 5.24 |
| MMA-4.0-sd1 | 98.81 | 61.88 | 52.64 | 93.98 | 73.70 | 37.78 | 5.11 |
| MMA-6.0-sd0 | 98.32 | 62.32 | 53.31 | 93.16 | 72.63 | 38.78 | 8.69 |
| MMA-6.0-sd1 | 98.50 | 62.49 | 53.48 | 93.48 | 73.50 | 38.63 | 8.32 |
| OMMA-2.0-sd0 | 99.26 | 54.01 | 42.69 | 95.94 | 67.78 | 7.03 | 0.03 |
| OMMA-2.0-sd1 | 99.21 | 54.04 | 42.74 | 95.72 | 68.83 | 6.42 | 0.00 |
| OMMA-4.0-sd0 | 98.61 | 62.17 | 53.06 | 94.06 | 73.51 | 39.66 | 5.02 |
| OMMA-4.0-sd1 | 98.61 | 62.01 | 52.86 | 93.72 | 73.18 | 38.98 | 5.58 |
| OMMA-6.0-sd0 | 98.16 | 62.45 | 53.52 | 92.90 | 72.59 | 39.68 | 8.93 |
| OMMA-6.0-sd1 | 98.45 | 62.24 | 53.19 | 93.37 | 72.93 | 37.63 | 8.83 |
| PGD-ens | 98.87 | 56.13 | 45.44 | 94.37 | 70.16 | 16.79 | 0.46 |
| PGDLS-ens | 99.14 | 54.71 | 43.60 | 94.52 | 67.45 | 12.33 | 0.11 |
| DDN-Rony et al. | 99.02 | 59.93 | 50.15 | 95.65 | 77.65 | 25.44 | 1.87 |

Table 7: Accuracies of models trained on CIFAR10 with $\ell_2$-norm constrained attacks. These robust accuracies are calculated under both combined (whitebox+transfer) PGD attacks. sd0 and sd1 indicate 2 different random seeds.

| CIFAR10 Model | Cln Acc | AvgAcc | AvgRobAcc | RobAcc under different $\epsilon$, combined (whitebox+transfer) attacks | | | | |
| --- | --- | --- | --- | --- | --- | --- | --- | --- |
| | | | | 0.5 | 1.0 | 1.5 | 2.0 | 2.5 |
| STD | 94.92 | 15.82 | 0.00 | 0.01 | 0.00 | 0.00 | 0.00 | 0.00 |
| PGD-0.5 | 89.10 | 33.63 | 22.53 | 65.61 | 33.21 | 11.25 | 2.31 | 0.28 |
| PGD-1.0 | 83.25 | 39.70 | 30.99 | 66.69 | 46.08 | 26.05 | 11.92 | 4.21 |
| PGD-1.5 | 75.80 | 41.75 | 34.94 | 62.70 | 48.32 | 33.72 | 20.07 | 9.91 |
| PGD-2.0 | 71.05 | 41.78 | 35.92 | 59.76 | 47.85 | 35.29 | 23.15 | 13.56 |
| PGD-2.5 | 65.17 | 40.93 | 36.08 | 55.60 | 45.76 | 35.76 | 26.00 | 17.27 |
| PGDLS-0.5 | 89.43 | 33.41 | 22.21 | 65.49 | 32.40 | 10.73 | 2.09 | 0.33 |
| PGDLS-1.0 | 83.62 | 39.46 | 30.63 | 67.29 | 45.30 | 25.43 | 11.08 | 4.03 |
| PGDLS-1.5 | 77.03 | 41.74 | 34.68 | 63.76 | 48.43 | 33.04 | 19.00 | 9.17 |
| PGDLS-2.0 | 72.14 | 42.15 | 36.16 | 60.90 | 48.22 | 35.21 | 23.19 | 13.26 |
| PGDLS-2.5 | 66.21 | 41.21 | 36.21 | 56.45 | 46.66 | 35.93 | 25.51 | 16.51 |
| MMA-1.0-sd0 | 88.02 | 35.55 | 25.06 | 66.18 | 37.75 | 15.58 | 4.74 | 1.03 |
| MMA-1.0-sd1 | 88.92 | 35.69 | 25.05 | 66.81 | 37.16 | 15.71 | 4.49 | 1.07 |
| MMA-2.0-sd0 | 84.22 | 40.48 | 31.73 | 65.91 | 45.66 | 27.40 | 14.18 | 5.50 |
| MMA-2.0-sd1 | 85.16 | 39.81 | 30.75 | 65.36 | 44.44 | 26.42 | 12.63 | 4.88 |
| MMA-3.0-sd0 | 82.11 | 41.59 | 33.49 | 64.22 | 46.41 | 30.23 | 17.85 | 8.73 |
| MMA-3.0-sd1 | 81.79 | 41.16 | 33.03 | 63.58 | 45.59 | 29.77 | 17.52 | 8.69 |
| OMMA-1.0-sd0 | 89.02 | 35.18 | 24.41 | 65.43 | 36.89 | 14.77 | 4.18 | 0.79 |
| OMMA-1.0-sd1 | 89.97 | 35.20 | 24.25 | 66.16 | 36.10 | 14.04 | 4.17 | 0.79 |
| OMMA-2.0-sd0 | 86.06 | 39.32 | 29.97 | 65.28 | 43.82 | 24.85 | 11.53 | 4.36 |
| OMMA-2.0-sd1 | 85.04 | 39.68 | 30.61 | 64.69 | 44.36 | 25.89 | 12.92 | 5.19 |
| OMMA-3.0-sd0 | 83.86 | 40.62 | 31.97 | 64.14 | 45.61 | 28.12 | 15.00 | 6.97 |
| OMMA-3.0-sd1 | 84.00 | 40.66 | 32.00 | 63.81 | 45.22 | 28.47 | 15.41 | 7.08 |
| PGD-ens | 85.63 | 40.39 | 31.34 | 62.98 | 45.87 | 27.91 | 14.23 | 5.72 |
| PGDLS-ens | 86.11 | 40.38 | 31.23 | 63.74 | 46.21 | 27.58 | 13.32 | 5.31 |
| DDN-Rony et al. | 89.05 | 36.23 | 25.67 | 66.51 | 39.02 | 16.60 | 5.02 | 1.20 |

Table 8: Accuracies of models trained on MNIST with $\ell_\infty$-norm constrained attacks. These robust accuracies are calculated under only whitebox PGD attacks. sd0 and sd1 indicate 2 different random seeds.

| MNIST Model | Cln Acc | AvgAcc | AvgRobAcc | RobAcc under different $\epsilon$, whitebox only | | | |
| --- | --- | --- | --- | --- | --- | --- | --- |
| | | | | 0.1 | 0.2 | 0.3 | 0.4 |
| STD | 99.21 | 35.02 | 18.97 | 73.59 | 2.31 | 0.00 | 0.00 |
| PGD-0.1 | 99.40 | 48.91 | 36.29 | 96.35 | 48.71 | 0.09 | 0.00 |
| PGD-0.2 | 99.22 | 57.93 | 47.60 | 97.44 | 92.12 | 0.86 | 0.00 |
| PGD-0.3 | 98.96 | 77.35 | 71.95 | 97.90 | 96.00 | 91.86 | 2.03 |
| PGD-0.4 | 96.64 | 91.51 | 90.22 | 94.79 | 92.27 | 88.82 | 85.02 |
| PGD-0.45 | 11.35 | 11.35 | 11.35 | 11.35 | 11.35 | 11.35 | 11.35 |
| PGDLS-0.1 | 99.43 | 46.94 | 33.82 | 95.41 | 39.85 | 0.02 | 0.00 |
| PGDLS-0.2 | 99.38 | 58.44 | 48.20 | 97.38 | 89.49 | 5.95 | 0.00 |
| PGDLS-0.3 | 99.10 | 76.85 | 71.29 | 97.98 | 95.66 | 90.63 | 0.90 |
| PGDLS-0.4 | 98.98 | 95.49 | 94.61 | 98.13 | 96.42 | 94.02 | 89.89 |
| PGDLS-0.45 | 98.89 | 95.72 | 94.92 | 97.91 | 96.64 | 94.54 | 90.60 |
| MMA-0.45-sd0 | 98.95 | 94.97 | 93.97 | 97.89 | 96.26 | 93.57 | 88.16 |
| MMA-0.45-sd1 | 98.90 | 94.83 | 93.81 | 97.83 | 96.18 | 93.34 | 87.91 |
| OMMA-0.45-sd0 | 98.98 | 95.06 | 94.07 | 97.91 | 96.22 | 93.63 | 88.54 |
| OMMA-0.45-sd1 | 99.02 | 95.45 | 94.55 | 97.96 | 96.30 | 94.16 | 89.80 |
| PGD-ens | 99.28 | 58.02 | 47.70 | 97.31 | 90.11 | 3.38 | 0.00 |
| PGDLS-ens | 99.34 | 59.09 | 49.02 | 97.50 | 90.56 | 8.03 | 0.00 |
| PGD-Madry et al. | 98.53 | 76.08 | 70.47 | 97.08 | 94.87 | 89.79 | 0.13 |
| TRADES | 99.47 | 77.95 | 72.57 | 98.81 | 97.27 | 94.13 | 0.07 |

Table 9: Accuracies of models trained on CIFAR10 with $\ell_\infty$-norm constrained attacks. These robust accuracies are calculated under only whitebox PGD attacks. sd0 and sd1 indicate 2 different random seeds.

| CIFAR10 Model | Cln Acc | AvgAcc | AvgRobAcc | RobAcc under different $\epsilon$, whitebox only | | | | | | | |
| --- | --- | --- | --- | --- | --- | --- | --- | --- | --- | --- | --- |
| | | | | 4 | 8 | 12 | 16 | 20 | 24 | 28 | 32 |
| STD | 94.92 | 10.55 | 0.00 | 0.00 | 0.00 | 0.00 | 0.00 | 0.00 | 0.00 | 0.00 | 0.00 |
| PGD-4 | 90.44 | 22.97 | 14.53 | 66.33 | 33.51 | 12.27 | 3.03 | 0.77 | 0.25 | 0.07 | 0.02 |
| PGD-8 | 85.14 | 27.28 | 20.05 | 67.73 | 46.49 | 26.69 | 12.37 | 4.71 | 1.58 | 0.62 | 0.23 |
| PGD-12 | 77.86 | 28.55 | 22.39 | 63.90 | 48.25 | 32.19 | 18.78 | 9.58 | 4.12 | 1.59 | 0.72 |
| PGD-16 | 68.86 | 28.42 | 23.36 | 58.07 | 46.17 | 33.84 | 22.99 | 13.65 | 7.19 | 3.43 | 1.57 |
| PGD-20 | 61.06 | 27.73 | 23.57 | 51.75 | 43.32 | 34.22 | 25.19 | 16.36 | 9.65 | 5.33 | 2.73 |
| PGD-24 | 10.90 | 9.98 | 9.86 | 10.60 | 10.34 | 10.11 | 10.01 | 9.91 | 9.74 | 9.39 | 8.81 |
| PGD-28 | 10.00 | 10.00 | 10.00 | 10.00 | 10.00 | 10.00 | 10.00 | 10.00 | 10.00 | 10.00 | 10.00 |
| PGD-32 | 10.00 | 10.00 | 10.00 | 10.00 | 10.00 | 10.00 | 10.00 | 10.00 | 10.00 | 10.00 | 10.00 |
| PGDLS-4 | 89.87 | 22.43 | 14.00 | 63.98 | 31.93 | 11.57 | 3.43 | 0.77 | 0.18 | 0.09 | 0.05 |
| PGDLS-8 | 85.63 | 27.22 | 19.92 | 67.96 | 46.19 | 26.24 | 12.28 | 4.54 | 1.52 | 0.45 | 0.21 |
| PGDLS-12 | 79.39 | 28.50 | 22.14 | 64.63 | 48.10 | 31.40 | 17.99 | 8.80 | 4.01 | 1.51 | 0.67 |
| PGDLS-16 | 70.68 | 28.53 | 23.26 | 59.44 | 47.04 | 33.78 | 21.94 | 12.79 | 6.66 | 3.07 | 1.34 |
| PGDLS-20 | 65.81 | 27.82 | 23.07 | 54.96 | 44.46 | 33.41 | 22.94 | 14.27 | 8.07 | 4.37 | 2.08 |
| PGDLS-24 | 58.36 | 27.25 | 23.36 | 49.09 | 41.47 | 32.90 | 24.84 | 16.93 | 10.88 | 7.04 | 3.76 |
| PGDLS-28 | 50.07 | 25.68 | 22.63 | 40.77 | 35.07 | 30.18 | 24.76 | 19.40 | 14.22 | 9.96 | 6.65 |
| PGDLS-32 | 38.80 | 22.79 | 20.79 | 26.19 | 25.34 | 24.72 | 23.21 | 20.98 | 18.13 | 15.12 | 12.66 |
| MMA-12-sd0 | 88.59 | 27.54 | 19.91 | 67.99 | 43.62 | 24.79 | 12.74 | 5.85 | 2.68 | 1.09 | 0.51 |
| MMA-12-sd1 | 88.91 | 26.68 | 18.90 | 67.17 | 43.63 | 23.62 | 10.80 | 4.07 | 1.20 | 0.50 | 0.18 |
| MMA-20-sd0 | 86.56 | 31.72 | 24.87 | 67.07 | 48.74 | 34.06 | 21.97 | 13.37 | 7.56 | 4.06 | 2.11 |
| MMA-20-sd1 | 85.87 | 33.07 | 26.47 | 65.63 | 48.11 | 34.70 | 24.73 | 16.45 | 10.97 | 7.00 | 4.14 |
| MMA-32-sd0 | 84.36 | 36.58 | 30.60 | 65.25 | 50.20 | 38.78 | 30.01 | 22.57 | 16.66 | 12.30 | 9.07 |
| MMA-32-sd1 | 84.76 | 33.49 | 27.08 | 64.66 | 48.23 | 35.65 | 25.74 | 17.86 | 11.86 | 7.79 | 4.88 |
| OMMA-12-sd0 | 88.52 | 29.34 | 21.94 | 67.49 | 46.11 | 29.22 | 16.65 | 8.62 | 4.36 | 2.05 | 1.03 |
| OMMA-12-sd1 | 87.82 | 30.30 | 23.11 | 66.77 | 46.77 | 31.19 | 19.40 | 10.93 | 5.72 | 2.84 | 1.29 |
| OMMA-20-sd0 | 87.06 | 36.00 | 29.61 | 68.00 | 52.98 | 40.13 | 28.92 | 19.78 | 13.04 | 8.47 | 5.60 |
| OMMA-20-sd1 | 87.44 | 34.49 | 27.87 | 67.40 | 51.55 | 37.94 | 26.48 | 17.76 | 11.31 | 6.74 | 3.76 |
| OMMA-32-sd0 | 86.11 | 38.87 | 32.97 | 67.57 | 53.70 | 42.56 | 32.88 | 24.91 | 18.57 | 13.79 | 9.76 |
| OMMA-32-sd1 | 86.36 | 39.13 | 33.23 | 68.80 | 56.02 | 44.62 | 33.97 | 24.71 | 17.37 | 11.94 | 8.39 |
| PGD-ens | 87.38 | 28.83 | 21.51 | 64.85 | 47.67 | 30.37 | 16.63 | 7.79 | 3.01 | 1.25 | 0.52 |
| PGDLS-ens | 76.73 | 30.60 | 24.83 | 61.16 | 49.46 | 36.63 | 23.90 | 13.92 | 7.62 | 3.91 | 2.05 |
| PGD-Madry et al. | 87.14 | 27.36 | 19.89 | 68.01 | 44.70 | 25.15 | 12.52 | 5.50 | 2.25 | 0.73 | 0.27 |
| TRADES | 84.92 | 30.46 | 23.66 | 70.96 | 52.92 | 33.04 | 18.26 | 8.37 | 3.59 | 1.42 | 0.69 |

Table 10: Accuracies of models trained on MNIST with $\ell_2$-norm constrained attacks. These robust accuracies are calculated under only whitebox PGD attacks. sd0 and sd1 indicate 2 different random seeds.

| MNIST Model | Cln Acc | AvgAcc | AvgRobAcc | RobAcc under different $\epsilon$, whitebox only | | | |
|---|---|---|---|---|---|---|---|
| | | | | 1.0 | 2.0 | 3.0 | 4.0 |
| STD | 99.21 | 41.90 | 27.57 | 86.61 | 23.02 | 0.64 | 0.00 |
| PGD-1.0 | 99.30 | 49.55 | 37.11 | 95.07 | 48.99 | 4.36 | 0.01 |
| PGD-2.0 | 98.76 | 56.38 | 45.79 | 94.82 | 72.94 | 15.08 | 0.31 |
| PGD-3.0 | 97.14 | 60.94 | 51.89 | 90.02 | 71.53 | 40.72 | 5.28 |
| PGD-4.0 | 93.41 | 59.93 | 51.56 | 82.41 | 66.49 | 44.36 | 12.99 |
| PGDLS-1.0 | 99.39 | 48.17 | 35.36 | 94.35 | 43.96 | 2.97 | 0.16 |
| PGDLS-2.0 | 99.09 | 55.17 | 44.19 | 95.22 | 69.73 | 11.80 | 0.03 |
| PGDLS-3.0 | 97.52 | 60.60 | 51.37 | 90.87 | 72.24 | 38.39 | 3.99 |
| PGDLS-4.0 | 93.68 | 59.89 | 51.44 | 82.73 | 67.37 | 44.59 | 11.07 |
| MMA-2.0-sd0 | 99.27 | 53.97 | 42.64 | 95.59 | 68.66 | 6.32 | 0.01 |
| MMA-2.0-sd1 | 99.28 | 54.46 | 43.26 | 95.79 | 68.45 | 8.79 | 0.01 |
| MMA-4.0-sd0 | 98.71 | 62.51 | 53.45 | 93.93 | 74.06 | 40.02 | 5.81 |
| MMA-4.0-sd1 | 98.81 | 62.22 | 53.07 | 93.98 | 73.81 | 38.76 | 5.75 |
| MMA-6.0-sd0 | 98.32 | 62.60 | 53.67 | 93.16 | 72.72 | 39.47 | 9.35 |
| MMA-6.0-sd1 | 98.50 | 62.73 | 53.79 | 93.48 | 73.57 | 39.25 | 8.86 |
| OMMA-2.0-sd0 | 99.26 | 54.12 | 42.83 | 95.94 | 68.08 | 7.27 | 0.03 |
| OMMA-2.0-sd1 | 99.21 | 54.12 | 42.85 | 95.72 | 68.96 | 6.72 | 0.00 |
| OMMA-4.0-sd0 | 98.61 | 62.44 | 53.40 | 94.06 | 73.60 | 40.29 | 5.66 |
| OMMA-4.0-sd1 | 98.61 | 62.22 | 53.13 | 93.72 | 73.23 | 39.53 | 6.03 |
| OMMA-6.0-sd0 | 98.16 | 62.67 | 53.79 | 92.90 | 72.71 | 40.28 | 9.29 |
| OMMA-6.0-sd1 | 98.45 | 62.52 | 53.54 | 93.37 | 73.02 | 38.49 | 9.28 |
| PGD-ens | 98.87 | 56.57 | 45.99 | 94.73 | 70.98 | 17.76 | 0.51 |
| PGDLS-ens | 99.14 | 54.98 | 43.93 | 94.86 | 68.08 | 12.68 | 0.12 |
| DDN-Rony et al. | 99.02 | 60.34 | 50.67 | 95.65 | 77.79 | 26.59 | 2.64 |

Table 11: Accuracies of models trained on CIFAR10 with $\ell_2$-norm constrained attacks. These robust accuracies are calculated under only whitebox PGD attacks. sd0 and sd1 indicate 2 different random seeds.

| CIFAR10 Model | Cln Acc | AvgAcc | AvgRobAcc | RobAcc under different $\epsilon$, whitebox only | | | | |
|---|---|---|---|---|---|---|---|---|
| | | | | 0.5 | 1.0 | 1.5 | 2.0 | 2.5 |
| STD | 94.92 | 15.82 | 0.00 | 0.01 | 0.00 | 0.00 | 0.00 | 0.00 |
| PGD-0.5 | 89.10 | 33.64 | 22.55 | 65.61 | 33.23 | 11.29 | 2.34 | 0.29 |
| PGD-1.0 | 83.25 | 39.74 | 31.04 | 66.69 | 46.11 | 26.16 | 12.00 | 4.26 |
| PGD-1.5 | 75.80 | 41.81 | 35.02 | 62.74 | 48.35 | 33.80 | 20.17 | 10.03 |
| PGD-2.0 | 71.05 | 41.88 | 36.05 | 59.80 | 47.92 | 35.39 | 23.34 | 13.81 |
| PGD-2.5 | 65.17 | 41.03 | 36.20 | 55.66 | 45.82 | 35.90 | 26.14 | 17.49 |
| PGDLS-0.5 | 89.43 | 33.44 | 22.25 | 65.50 | 32.42 | 10.78 | 2.17 | 0.36 |
| PGDLS-1.0 | 83.62 | 39.50 | 30.68 | 67.30 | 45.35 | 25.49 | 11.19 | 4.08 |
| PGDLS-1.5 | 77.03 | 41.80 | 34.75 | 63.76 | 48.46 | 33.11 | 19.12 | 9.32 |
| PGDLS-2.0 | 72.14 | 42.24 | 36.27 | 60.96 | 48.28 | 35.32 | 23.38 | 13.39 |
| PGDLS-2.5 | 66.21 | 41.34 | 36.36 | 56.49 | 46.72 | 36.13 | 25.73 | 16.75 |
| MMA-1.0-sd0 | 88.02 | 35.58 | 25.09 | 66.19 | 37.80 | 15.61 | 4.79 | 1.06 |
| MMA-1.0-sd1 | 88.92 | 35.74 | 25.10 | 66.81 | 37.22 | 15.78 | 4.57 | 1.14 |
| MMA-2.0-sd0 | 84.22 | 41.22 | 32.62 | 65.98 | 46.11 | 28.56 | 15.60 | 6.86 |
| MMA-2.0-sd1 | 85.16 | 40.60 | 31.69 | 65.45 | 45.27 | 28.07 | 13.99 | 5.67 |
| MMA-3.0-sd0 | 82.11 | 43.67 | 35.98 | 64.25 | 47.61 | 33.48 | 22.07 | 12.50 |
| MMA-3.0-sd1 | 81.79 | 43.75 | 36.14 | 63.82 | 47.33 | 33.79 | 22.36 | 13.40 |
| OMMA-1.0-sd0 | 89.02 | 35.49 | 24.79 | 65.46 | 37.38 | 15.34 | 4.76 | 1.00 |
| OMMA-1.0-sd1 | 89.97 | 35.41 | 24.49 | 66.24 | 36.47 | 14.44 | 4.43 | 0.89 |
| OMMA-2.0-sd0 | 86.06 | 42.80 | 34.14 | 65.55 | 46.29 | 30.60 | 18.23 | 10.05 |
| OMMA-2.0-sd1 | 85.04 | 42.96 | 34.55 | 65.23 | 46.32 | 31.07 | 19.36 | 10.75 |
| OMMA-3.0-sd0 | 83.86 | 46.46 | 38.99 | 64.67 | 49.34 | 36.40 | 26.50 | 18.02 |
| OMMA-3.0-sd1 | 84.00 | 45.59 | 37.91 | 64.31 | 48.50 | 35.92 | 24.81 | 16.03 |
| PGD-ens | 85.63 | 41.32 | 32.46 | 63.27 | 46.66 | 29.35 | 15.95 | 7.09 |
| PGDLS-ens | 86.11 | 41.39 | 32.45 | 64.04 | 46.99 | 29.11 | 15.51 | 6.59 |
| DDN-Rony et al. | 89.05 | 36.25 | 25.69 | 66.51 | 39.02 | 16.63 | 5.05 | 1.24 |

Table 12: The TransferGap of models trained on MNIST with $\ell_\infty$-norm constrained attacks. TransferGap indicates the gap between robust accuracy under only whitebox PGD attacks and under combined (whitebox+transfer) PGD attacks. sd0 and sd1 indicate 2 different random seeds.

| MNIST Model | Cln Acc | AvgAcc | AvgRobAcc | TransferGap: RobAcc drop after adding transfer attacks | | | |
|---|---|---|---|---|---|---|---|
| | | | | 0.1 | 0.2 | 0.3 | 0.4 |
| STD | - | 0.00 | 0.00 | 0.01 | 0.00 | 0.00 | 0.00 |
| PGD-0.1 | - | 0.06 | 0.07 | 0.00 | 0.20 | 0.08 | 0.00 |
| PGD-0.2 | - | 0.00 | 0.00 | 0.00 | 0.00 | 0.02 | 0.00 |
| PGD-0.3 | - | 0.38 | 0.48 | 0.00 | 0.00 | 0.10 | 1.81 |
| PGD-0.4 | - | 2.14 | 2.67 | 0.10 | 0.70 | 2.33 | 7.55 |
| PGD-0.45 | - | 0.00 | 0.00 | 0.00 | 0.00 | 0.00 | 0.00 |
| PGDLS-0.1 | - | 0.09 | 0.11 | 0.00 | 0.43 | 0.02 | 0.00 |
| PGDLS-0.2 | - | 0.08 | 0.11 | 0.00 | 0.00 | 0.42 | 0.00 |
| PGDLS-0.3 | - | 0.29 | 0.36 | 0.01 | 0.00 | 0.54 | 0.90 |
| PGDLS-0.4 | - | 2.42 | 3.02 | 0.01 | 0.13 | 1.01 | 10.93 |
| PGDLS-0.45 | - | 0.97 | 1.22 | 0.00 | 0.30 | 1.25 | 3.32 |
| MMA-0.45-sd0 | - | 0.83 | 1.04 | 0.02 | 0.25 | 0.98 | 2.92 |
| MMA-0.45-sd1 | - | 0.80 | 0.99 | 0.01 | 0.18 | 0.71 | 3.08 |
| OMMA-0.45-sd0 | - | 1.12 | 1.40 | 0.01 | 0.17 | 1.28 | 4.13 |
| OMMA-0.45-sd1 | - | 1.42 | 1.78 | 0.03 | 0.28 | 1.72 | 5.07 |
| PGD-ens | - | 0.04 | 0.05 | 0.06 | 0.12 | 0.01 | 0.00 |
| PGDLS-ens | - | 0.05 | 0.06 | 0.02 | 0.16 | 0.07 | 0.00 |
| PGD-Madry et al. | - | 0.04 | 0.05 | 0.00 | 0.04 | 0.15 | 0.02 |
| TRADES | 0.0 | 0.01 | 0.0 | 0.0 | 0.0 | 0.01 | 0.02 |

Table 13: The TransferGap of models trained on CIFAR10 with $\ell_\infty$-norm constrained attacks. TransferGap indicates the gap between robust accuracy under only whitebox PGD attacks and under combined (whitebox+transfer) PGD attacks. sd0 and sd1 indicate 2 different random seeds.

| CIFAR10 Model | Cln Acc | AvgAcc | AvgRobAcc | TransferGap: RobAcc drop after adding transfer attacks | | | | | | | |
|---|---|---|---|---|---|---|---|---|---|---|---|
| | | | | 4 | 8 | 12 | 16 | 20 | 24 | 28 | 32 |
| STD | - | 0.00 | 0.00 | 0.00 | 0.00 | 0.00 | 0.00 | 0.00 | 0.00 | 0.00 | 0.00 |
| PGD-4 | - | 0.02 | 0.02 | 0.02 | 0.02 | 0.05 | 0.02 | 0.02 | 0.01 | 0.01 | 0.01 |
| PGD-8 | - | 0.02 | 0.02 | 0.00 | 0.02 | 0.06 | 0.04 | 0.02 | 0.02 | 0.00 | 0.01 |
| PGD-12 | - | 0.05 | 0.05 | 0.02 | 0.03 | 0.06 | 0.11 | 0.10 | 0.07 | 0.03 | 0.02 |
| PGD-16 | - | 0.14 | 0.15 | 0.08 | 0.08 | 0.20 | 0.26 | 0.28 | 0.18 | 0.11 | 0.03 |
| PGD-20 | - | 0.39 | 0.44 | 0.03 | 0.19 | 0.49 | 0.64 | 0.70 | 0.60 | 0.59 | 0.31 |
| PGD-24 | - | 0.03 | 0.03 | 0.00 | 0.00 | 0.00 | 0.01 | 0.02 | 0.05 | 0.05 | 0.13 |
| PGD-28 | - | 0.00 | 0.00 | 0.00 | 0.00 | 0.00 | 0.00 | 0.00 | 0.00 | 0.00 | 0.00 |
| PGD-32 | - | 0.00 | 0.00 | 0.00 | 0.00 | 0.00 | 0.00 | 0.00 | 0.00 | 0.00 | 0.00 |
| PGDLS-4 | - | 0.04 | 0.04 | 0.00 | 0.01 | 0.10 | 0.11 | 0.09 | 0.02 | 0.01 | 0.00 |
| PGDLS-8 | - | 0.02 | 0.02 | 0.00 | 0.00 | 0.05 | 0.06 | 0.03 | 0.04 | 0.01 | 0.00 |
| PGDLS-12 | - | 0.05 | 0.05 | 0.01 | 0.02 | 0.06 | 0.13 | 0.11 | 0.06 | 0.03 | 0.02 |
| PGDLS-16 | - | 0.09 | 0.10 | 0.01 | 0.04 | 0.14 | 0.22 | 0.13 | 0.12 | 0.09 | 0.03 |
| PGDLS-20 | - | 0.21 | 0.24 | 0.00 | 0.07 | 0.28 | 0.41 | 0.47 | 0.28 | 0.29 | 0.13 |
| PGDLS-24 | - | 0.73 | 0.82 | 0.04 | 0.34 | 0.80 | 1.08 | 1.23 | 1.22 | 1.18 | 0.65 |
| PGDLS-28 | - | 1.47 | 1.66 | 0.06 | 0.46 | 1.18 | 1.99 | 2.57 | 2.73 | 2.34 | 1.92 |
| PGDLS-32 | - | 2.91 | 3.28 | 0.03 | 0.38 | 1.50 | 3.25 | 4.76 | 5.21 | 5.30 | 5.78 |
| MMA-12-sd0 | - | 0.67 | 0.76 | 0.03 | 0.20 | 0.72 | 1.29 | 1.58 | 1.25 | 0.64 | 0.35 |
| MMA-12-sd1 | - | 0.45 | 0.50 | 0.09 | 0.66 | 1.05 | 1.04 | 0.70 | 0.28 | 0.15 | 0.06 |
| MMA-20-sd0 | - | 2.86 | 3.22 | 0.15 | 1.85 | 4.23 | 5.42 | 5.23 | 4.31 | 2.89 | 1.68 |
| MMA-20-sd1 | - | 4.35 | 4.90 | 0.19 | 2.00 | 4.74 | 7.43 | 8.18 | 7.37 | 5.67 | 3.58 |
| MMA-32-sd0 | - | 7.19 | 8.09 | 0.43 | 3.02 | 7.29 | 11.10 | 12.41 | 11.89 | 10.33 | 8.26 |
| MMA-32-sd1 | - | 4.42 | 4.97 | 0.25 | 2.28 | 5.29 | 7.50 | 8.01 | 6.87 | 5.59 | 3.96 |
| OMMA-12-sd0 | - | 3.02 | 3.40 | 0.53 | 3.53 | 6.00 | 6.36 | 5.19 | 3.12 | 1.59 | 0.90 |
| OMMA-12-sd1 | - | 4.06 | 4.57 | 0.54 | 3.67 | 7.62 | 9.08 | 7.37 | 4.68 | 2.46 | 1.15 |
| OMMA-20-sd0 | - | 8.59 | 9.66 | 1.46 | 7.59 | 13.84 | 15.83 | 14.46 | 11.08 | 7.68 | 5.37 |
| OMMA-20-sd1 | - | 6.72 | 7.56 | 1.12 | 5.95 | 10.61 | 12.48 | 11.72 | 9.08 | 6.00 | 3.51 |
| OMMA-32-sd0 | - | 10.51 | 11.83 | 1.55 | 7.39 | 13.68 | 16.90 | 17.47 | 15.63 | 12.67 | 9.31 |
| OMMA-32-sd1 | - | 10.38 | 11.67 | 1.94 | 8.90 | 14.99 | 17.88 | 17.15 | 13.99 | 10.63 | 7.92 |
| PGD-ens | - | 0.73 | 0.82 | 0.26 | 0.72 | 1.49 | 1.53 | 1.44 | 0.66 | 0.34 | 0.13 |
| PGDLS-ens | - | 1.08 | 1.21 | 0.64 | 1.25 | 1.57 | 1.76 | 1.64 | 1.45 | 0.77 | 0.62 |
| PGD-Madry et al. | - | 0.14 | 0.16 | 0.00 | 0.02 | 0.12 | 0.37 | 0.32 | 0.30 | 0.09 | 0.04 |
| TRADES | 0.0 | 0.0 | 0.01 | 0.0 | 0.0 | 0.0 | 0.03 | 0.02 | 0.02 | 0.02 | 0.0 |

Table 14: The TransferGap of models trained on MNIST with $\ell_2$-norm constrained attacks. TransferGap indicates the gap between robust accuracy under only whitebox PGD attacks and under combined (whitebox+transfer) PGD attacks. sd0 and sd1 indicate 2 different random seeds.

| MNIST Model | Cln Acc | AvgAcc | AvgRobAcc | TransferGap: RobAcc drop after adding transfer attacks | | | |
|---|---|---|---|---|---|---|---|
| | | | | 1.0 | 2.0 | 3.0 | 4.0 |
| STD | - | 0.06 | 0.07 | 0.00 | 0.24 | 0.05 | 0.00 |
| PGD-1.0 | - | 0.76 | 0.96 | 0.01 | 2.15 | 1.65 | 0.01 |
| PGD-2.0 | - | 0.24 | 0.30 | 0.00 | 0.24 | 0.88 | 0.10 |
| PGD-3.0 | - | 0.57 | 0.72 | 0.01 | 0.50 | 1.79 | 0.57 |
| PGD-4.0 | - | 0.41 | 0.51 | 0.07 | 0.24 | 0.92 | 0.81 |
| PGDLS-1.0 | - | 0.56 | 0.70 | 0.02 | 1.52 | 1.08 | 0.16 |
| PGDLS-2.0 | - | 0.44 | 0.55 | 0.00 | 0.40 | 1.79 | 0.02 |
| PGDLS-3.0 | - | 0.47 | 0.59 | 0.01 | 0.33 | 1.59 | 0.43 |
| PGDLS-4.0 | - | 0.39 | 0.49 | 0.06 | 0.16 | 0.91 | 0.84 |
| MMA-2.0-sd0 | - | 0.12 | 0.15 | 0.00 | 0.29 | 0.29 | 0.00 |
| MMA-2.0-sd1 | - | 0.13 | 0.16 | 0.01 | 0.27 | 0.34 | 0.01 |
| MMA-4.0-sd0 | - | 0.26 | 0.33 | 0.00 | 0.05 | 0.68 | 0.57 |
| MMA-4.0-sd1 | - | 0.35 | 0.43 | 0.00 | 0.11 | 0.98 | 0.64 |
| MMA-6.0-sd0 | - | 0.29 | 0.36 | 0.00 | 0.09 | 0.69 | 0.66 |
| MMA-6.0-sd1 | - | 0.25 | 0.31 | 0.00 | 0.07 | 0.62 | 0.54 |
| OMMA-2.0-sd0 | - | 0.11 | 0.13 | 0.00 | 0.30 | 0.24 | 0.00 |
| OMMA-2.0-sd1 | - | 0.09 | 0.11 | 0.00 | 0.13 | 0.30 | 0.00 |
| OMMA-4.0-sd0 | - | 0.27 | 0.34 | 0.00 | 0.09 | 0.63 | 0.64 |
| OMMA-4.0-sd1 | - | 0.21 | 0.26 | 0.00 | 0.05 | 0.55 | 0.45 |
| OMMA-6.0-sd0 | - | 0.22 | 0.27 | 0.00 | 0.12 | 0.60 | 0.36 |
| OMMA-6.0-sd1 | - | 0.28 | 0.35 | 0.00 | 0.09 | 0.86 | 0.45 |
| PGD-ens | - | 0.44 | 0.55 | 0.36 | 0.82 | 0.97 | 0.05 |
| PGDLS-ens | - | 0.27 | 0.33 | 0.34 | 0.63 | 0.35 | 0.01 |
| DDN-Rony et al. | - | 0.41 | 0.51 | 0.00 | 0.14 | 1.15 | 0.77 |

Table 15: The TransferGap of models trained on CIFAR10 with $\ell_2$-norm constrained attacks. TransferGap indicates the gap between robust accuracy under only whitebox PGD attacks and under combined (whitebox+transfer) PGD attacks. sd0 and sd1 indicate 2 different random seeds.

| CIFAR10 Model | Cln Acc | AvgAcc | AvgRobAcc | TransferGap: RobAcc drop after adding transfer attacks | | | | |
|---|---|---|---|---|---|---|---|---|
| | | | | 0.5 | 1.0 | 1.5 | 2.0 | 2.5 |
| STD | - | 0.00 | 0.00 | 0.00 | 0.00 | 0.00 | 0.00 | 0.00 |
| PGD-0.5 | - | 0.02 | 0.02 | 0.00 | 0.02 | 0.04 | 0.03 | 0.01 |
| PGD-1.0 | - | 0.04 | 0.05 | 0.00 | 0.03 | 0.11 | 0.08 | 0.05 |
| PGD-1.5 | - | 0.06 | 0.07 | 0.04 | 0.03 | 0.08 | 0.10 | 0.12 |
| PGD-2.0 | - | 0.11 | 0.13 | 0.04 | 0.07 | 0.10 | 0.19 | 0.25 |
| PGD-2.5 | - | 0.10 | 0.12 | 0.06 | 0.06 | 0.14 | 0.14 | 0.22 |
| PGDLS-0.5 | - | 0.03 | 0.04 | 0.01 | 0.02 | 0.05 | 0.08 | 0.03 |
| PGDLS-1.0 | - | 0.05 | 0.06 | 0.01 | 0.05 | 0.06 | 0.11 | 0.05 |
| PGDLS-1.5 | - | 0.06 | 0.07 | 0.00 | 0.03 | 0.07 | 0.12 | 0.15 |
| PGDLS-2.0 | - | 0.09 | 0.11 | 0.06 | 0.06 | 0.11 | 0.19 | 0.13 |
| PGDLS-2.5 | - | 0.13 | 0.15 | 0.04 | 0.06 | 0.20 | 0.22 | 0.24 |
| MMA-1.0-sd0 | - | 0.03 | 0.03 | 0.01 | 0.05 | 0.03 | 0.05 | 0.03 |
| MMA-1.0-sd1 | - | 0.05 | 0.06 | 0.00 | 0.06 | 0.07 | 0.08 | 0.07 |
| MMA-2.0-sd0 | - | 0.74 | 0.89 | 0.07 | 0.45 | 1.16 | 1.42 | 1.36 |
| MMA-2.0-sd1 | - | 0.79 | 0.94 | 0.09 | 0.83 | 1.65 | 1.36 | 0.79 |
| MMA-3.0-sd0 | - | 2.08 | 2.49 | 0.03 | 1.20 | 3.25 | 4.22 | 3.77 |
| MMA-3.0-sd1 | - | 2.59 | 3.11 | 0.24 | 1.74 | 4.02 | 4.84 | 4.71 |
| OMMA-1.0-sd0 | - | 0.31 | 0.38 | 0.03 | 0.49 | 0.57 | 0.58 | 0.21 |
| OMMA-1.0-sd1 | - | 0.20 | 0.24 | 0.08 | 0.37 | 0.40 | 0.26 | 0.10 |
| OMMA-2.0-sd0 | - | 3.48 | 4.18 | 0.27 | 2.47 | 5.75 | 6.70 | 5.69 |
| OMMA-2.0-sd1 | - | 3.28 | 3.94 | 0.54 | 1.96 | 5.18 | 6.44 | 5.56 |
| OMMA-3.0-sd0 | - | 5.85 | 7.02 | 0.53 | 3.73 | 8.28 | 11.50 | 11.05 |
| OMMA-3.0-sd1 | - | 4.93 | 5.92 | 0.50 | 3.28 | 7.45 | 9.40 | 8.95 |
| PGD-ens | - | 0.94 | 1.12 | 0.29 | 0.79 | 1.44 | 1.72 | 1.37 |
| PGDLS-ens | - | 1.01 | 1.22 | 0.30 | 0.78 | 1.53 | 2.19 | 1.28 |
| DDN-Rony et al. | - | 0.02 | 0.02 | 0.00 | 0.00 | 0.03 | 0.03 | 0.04 |

