# OpenReview forum: "MMA Training: Direct Input Space Margin Maximization through Adversarial Training"
_ICLR.cc/2020/Conference — Accept (Poster)_

### Official Review · AnonReviewer3 · 2019-10-13
**Official Blind Review #3**

**Rating:** 3

**Review:**

Summary:
This paper proposes an adaptive margin-based adversarial training (eg. MMA) approach to train robust DNNs by maximizing the shortest margin of inputs to the decision boundary. Theoretical analyses have been provided to understand the connection between robust optimization and margin maximization. The main difference between the proposed approach to standard adversarial training is the adaptive selection of the perturbation bound \epsilon. This makes adversarial training with large perturbation possible, which was previously unachievable by standard adversarial training (Madry et al.) Empirical results match the theoretical analysis.

Pros:
1. The margin maximization idea has been well-explained, both intuitively and theoretically.
2. Interesting theoretical analyses and understandings of robust optimization from the margin perspective.
3. Clear advantage of MMA over standard adversarial training under large perturbations.

8. The proposed PGDLS is very interesting, actually quite good and much simpler, without extra computational cost. A similar idea was discussed in paper [3], where they gradually increase the convergence quality of training adversarial examples, and show the convergence guarantee  of "dynamic training".
9. The gradient-free SPSA helps confirm the improvements of MMA under large perturbations are not a side effect of gradient masking.


Cons:
1. The idea of "shortest successful perturbation" appears like a type of weak training attack, looking for minimum perturbations to just cross the classification boundary, like deepfool [1] or confidence 0 CW-L2 attack [2].
2. The margin d_\theta in Equation (1)/(2)/... defined on which norm? L_\infty or L2 norm? I assume it's the infinity norm. In Theorem 2.1, the \delta^{*} = argmin ||\delta||, is a norm? Looks like a mistake.
3. The minimum margin \delta^{*} is a bit confusing, is it used in maximization or just in the outer minimization? The last paragraph of page 3, L(\theta, \delta) or L(\delta, \theta), consistency check?
4. Why do we need the "gradients of margins to model parameters" analysis from Proposition 2.1 to remark 2.2? Given the \delta^{*} found in the inner maximization (eg. attacking) process (step 1), minimizing the loss over this \delta^{*}  seems quite a straightforward step 2. Why don't go directly from Theorem 2.1 to Proposition 2.4, since the extensions from LM loss to SLM and CE loss via Proposition 2.3 -> Proposition 2.4., just proves that the standard classification loss CE can already maximize the margin given \delta^{*}?
5. Section 4 Experiments. The experimental settings are not clear, and are not standard. What CIFAR10-\ell_{\infty} means: is it the CW-L2 attack, used for training, or for testing? How the test attacks were generated, the m and N, are confusing: for each test image, you have 260 samples for CIFAR10 (which means 260*10K in total), or just 260 in total (this is far less than a typical setting causing inaccurate results)? How are the d_max determined, and what are their relationship to standard \epsilon? How the m models were trained?
6. Fairness of the comparison. Since MMA changes \epsilon, how to fairly compare the robustness to standard epsilon bounded adversarial training is not discussed. Is it fair to compare MMA-3.0 vs PGD-2.5, since they have different epsilon? Why robustness was not tested against strong, unrestricted attacks like CW-L2 [2], and report the average L2 perturbations required to completely break the robustly trained model (and show MMA-trained models enforce large perturbations to succeed)?
7. Significance of the results. Normally, \epsilon_{infty} > 16/255 will cause perceptual difference. Under 16/255, PGD-8/16, PGDLS-8/16 are still the best. At this level, it is quite a surprise that MMA does not improve robustness, although it does increase clean accuracy. This means the theoretical analysis only stand under certain circumstances. I don't think the optimal \epsilon < margin can explain this, as it does not make sense to me the margin can be larger than 16/255. On the other hand, I thought the theoretical parts were discussing the  ROBUSTNESS, not the CLEAN ACCURACY? But it turns out the MMA benefits a lot the clean accuracy?  Why do we need robustness against large \infty perturbations, this definitely deserves more discussion, as when perturbation goes large, the L2 attack (eg. CW-L2) makes more sense than PGD-\infty.


[1] Moosavi-Dezfooli, Seyed-Mohsen, Alhussein Fawzi, and Pascal Frossard. "Deepfool: a simple and accurate method to fool deep neural networks." Proceedings of the IEEE conference on computer vision and pattern recognition. 2016.
[2] Carlini, Nicholas, and David Wagner. "Towards evaluating the robustness of neural networks." 2017 IEEE Symposium on Security and Privacy (SP). IEEE, 2017.
[3] Wang, Yisen, et al. "On the Convergence and Robustness of Adversarial Training." International Conference on Machine Learning. 2019.

============
My rating stays the same after reading through all the responses. I appreciate the authors' clarification on the notations and experimental settings. My 8/9 are positive points. My major concern is still the effectiveness of the proposed approach, and fairness of the comparison.  It seems that MMA only works when the perturbation is large, which often larger than the \epsilon used to train baseline adversarial training methods such as Trades. The authors seem have misunderstood my request for CWL2 results, I was just suggesting that the average L2 perturbation of CWL2 attack can be used as a fair test measure for robustness, instead of the AvgRobAcc used in the paper, and the susceptible comparison between  MMA-12 vs PGD-8, or MMA-32 vs Trades.

**Experience Assessment:**

I have published one or two papers in this area.

**Review Assessment: Checking Correctness Of Derivations And Theory:**

I carefully checked the derivations and theory.

**Review Assessment: Checking Correctness Of Experiments:**

I carefully checked the experiments.

**Review Assessment: Thoroughness In Paper Reading:**

I read the paper thoroughly.

---

> ### Author Response · Authors · 2019-11-11
> **Response to AnonReviewer3, part 1**
>
> Thank you for your detailed reviews and also valuing our contributions.
>
> We would like to carify with R3 (and other readers) that many items under "Cons" are not strictly "disadvantages" of our method. Most of them seem to be clarification questions, especially that item 8 and 9 seem to be comments with positive sentiment.
>
> Please let us know if any of our answers does not resolve the confusion, and we are happy to further elaborate.
>
>
> >> 1. "shortest successful perturbation" appears like a type of weak training attack, and similar to deepfool [1] or confidence 0 CW-L2 attack [2].
>
> You are right that the "shortest successful perturbation" is similar to deepfool and CW-L2. More precisely, deepfool, CW-L2 and our proposed AN-PGD are all algorithms for approximating the "shortest successful perturbation", as we mentioned in Section 2.3 "Other attacks that can serve a similar purpose can also fit into our MMA training framework".
> Specifically, the deepfool attack is not strong enough (e.g. as shown in Rony et al. 2019), probably due to it only uses first order approximation to find $\delta^*$, and therefore is not suitable for MMA training.
> On the other hand, CW-L2 is likely strong enough, but too expensive to compute during training.
>
> Moreover, while the "shortest successful perturbation", $\delta^*$, is a "weak" training attack if measured in the adversarial loss, it is the **right** attacks fo training. Attacks with magnitude larger than the margin $\|\delta^*\|$ could be stronger than $\delta^*$, our margin maximization theory suggests that training on "longer" (and thus stronger) perturbations does not necessarily increase the margin (Section 3 and Figure 2).
>
> Reference:
> Rony et al. Decoupling Direction and Norm for Efficient Gradient-Based L2 Adversarial Attacks and Defenses, CVPR 2019
>
> Please let us know if we do not directly address your concern out of this comment.
>
>
> >> 6. Fairness of comparison
>
> Please see response to all reviewers. Also please let us know if we do not fully address your concern, and we are happy to further elaborate on this.
>
>
> >> 6. Results on strong, unrestricted attacks like CW-L2
>
> We reported AvgRobAcc, the average robust accuracy over different perturbation magnitudes, including those with very large magnitudes. Therefore, AvgRobAcc serves similar purpose to average norm of the strong unrestricted attacks.
>
> We are working on experiments using CW-L2 to test our models trained with $\ell_2$ attacks, and will report the results when ready.
>
>
> >> 7. Why MMA does not improve robustness, but improves clean accuracy? This means the theoretical analysis only stand under certain circumstances.
>
> This result does not contradict our theory. In contrast, it is very well aligned with our theory.
> For wrongly classified examples, MMA training focuses on getting them classified correctly. For correctly classified examples, our theory suggests that MMA training tries to enlarge the margins of all of them, based on their intrinsic robustness (i.e. how difficult for a model to achieve a large margin on different points may be different).
> On the other hand, PGD training fails to adapt to the intrinsic robustness of different points, and thus significantly sacrifices its clean accuracy in order to achieve the slight additional robustness for large perturbation. This observation also echoes the sensitivity of PGD to its fixed (and arbitrary) perturbation magnitude. We will make this argument more clear in the paper.
>
> Please let us know if we need to further clarify.
>
>
> >> 7. Why do we need robustness against large $\ell_\infty$ perturbations? when perturbation goes large, the L2 attack (eg. CW-L2) makes more sense than PGD-$\ell_\infty$.
>
> We agree with R3 that perturbations that cause perceptual differences shall not be included to test the robustness of the model. However, it is hard to determine the boundary of "perceptual differences" in terms of the perturbation magnitude.
> Compared to PGD training, MMA provides a natural way in dealing with this dilemma: user can set $d_\max$ represents the magnitude that is "too large". Below $d_\max$, MMA training enlarges the margin of each individual example based on its robustness under the current model, to the maximium capacity of the model. In contrast, the fixed $\epsilon$ in PGD training need to be "large enough but not too large", which is much harder or even impossible to set, since each example could have different intrinsic robustness. As a result, MMA training is fairly insensitive to $d_\max$, but PGD training is very sensitive to $\epsilon$.
>
> In terms of what norm to measure the perturbation magnitude, we believe that it is more reasonable to evaluate the model using the norm that the model is trained on, namely "train on $\ell_2$ test on $\ell_2$", and "train on $\ell_\infty$ test on $\ell_\infty$". We will add CW-L2 results to models trained with $\ell_2$ attacks when they are ready.

---

> > ### Author Response · Authors · 2019-11-15
> > **Response to AnonReviewer3 on CW-L2 results**
> >
> > We run CW-L2 attack on the first 1000 examples (due to constraints on computational resources) for models trained with $\ell_2$ attacks. Therefore under each model, we have the minimum distance that CW-L2 needs (to make the prediction wrong) for each example.
> >
> > For each model, we show the clean accuracy, also mean, median, 25th percentile and 75th percentile of these minimum distances in the table below.
> >
> > Looking at CW-L2 results, we have very similar observations as compared to the observations we made in the paper (Appendix F) based on robust accuracies at different $\epsilon$'s and the AvgRobAcc (average robust accuracy).
> >
> > ----------
> > **MNIST models trained with $\ell_2$ attacks**
> >
> > STD              acc: 99.0% mean: 1.57, 25th: 1.19, median: 1.57, 75th: 1.95
> >
> > PGD-1.0          acc: 99.2% mean: 1.95, 25th: 1.58, median: 1.95, 75th: 2.33
> > PGD-2.0          acc: 98.6% mean: 2.32, 25th: 1.85, median: 2.43, 75th: 2.82
> > PGD-3.0          acc: 96.8% mean: 2.54, 25th: 1.79, median: 2.76, 75th: 3.39
> > PGD-4.0          acc: 91.8% mean: 2.53, 25th: 1.38, median: 2.75, 75th: 3.75
> >
> > PGDLS-1.0        acc: 99.3% mean: 1.88, 25th: 1.53, median: 1.88, 75th: 2.24
> > PGDLS-2.0        acc: 99.3% mean: 2.23, 25th: 1.82, median: 2.30, 75th: 2.68
> > PGDLS-3.0        acc: 96.9% mean: 2.51, 25th: 1.81, median: 2.74, 75th: 3.29
> > PGDLS-4.0        acc: 91.8% mean: 2.53, 25th: 1.46, median: 2.79, 75th: 3.72
> >
> > MMA-2.0          acc: 99.3% mean: 2.15, 25th: 1.81, median: 2.22, 75th: 2.58
> > MMA-4.0          acc: 98.5% mean: 2.67, 25th: 1.86, median: 2.72, 75th: 3.52
> > MMA-6.0          acc: 97.7% mean: 2.76, 25th: 1.84, median: 2.72, 75th: 3.61
> >
> > ----------
> >
> > On MNIST, for MMA trained models as $d_\max$ increases from 2.0 to 6.0, the mean minimum distance goes from 2.15 to 2.76, with the clean accuracy only drops to 97.7% from 99.3%. On the contrary, when we perform PGD/PGDLS with a larger $\epsilon$, e.g. $\epsilon=4.0$, the clean accuracy drops significantly to 91.8%, with the mean minimum distance increased to 2.53. Therefore MMA training achieves better performance on both clean accuracy and robustness under the MNIST-$\ell_2$ case.
> >
> >
> > ----------
> > **CIFAR10 models trained with $\ell_2$ attacks**
> >
> > STD              acc: 95.1%, mean: 0.09, 25th: 0.05, median: 0.08, 75th: 0.13
> >
> > PGD-0.5          acc: 89.2%, mean: 0.80, 25th: 0.32, median: 0.76, 75th: 1.21
> > PGD-1.0          acc: 83.8%, mean: 1.01, 25th: 0.30, median: 0.91, 75th: 1.62
> > PGD-1.5          acc: 76.0%, mean: 1.11, 25th: 0.04, median: 0.96, 75th: 1.87
> > PGD-2.0          acc: 71.6%, mean: 1.16, 25th: 0.00, median: 0.96, 75th: 2.03
> > PGD-2.5          acc: 65.2%, mean: 1.19, 25th: 0.00, median: 0.88, 75th: 2.10
> >
> > PGDLS-0.5        acc: 90.5%, mean: 0.79, 25th: 0.34, median: 0.76, 75th: 1.17
> > PGDLS-1.0        acc: 83.8%, mean: 1.00, 25th: 0.30, median: 0.91, 75th: 1.59
> > PGDLS-1.5        acc: 77.6%, mean: 1.10, 25th: 0.11, median: 0.97, 75th: 1.83
> > PGDLS-2.0        acc: 73.1%, mean: 1.16, 25th: 0.00, median: 0.98, 75th: 2.02
> > PGDLS-2.5        acc: 66.0%, mean: 1.19, 25th: 0.00, median: 0.88, 75th: 2.13
> >
> > MMA-1.0          acc: 88.4%, mean: 0.85, 25th: 0.32, median: 0.80, 75th: 1.27
> > MMA-2.0          acc: 84.2%, mean: 1.03, 25th: 0.26, median: 0.92, 75th: 1.62
> > MMA-3.0          acc: 81.2%, mean: 1.09, 25th: 0.21, median: 0.92, 75th: 1.73
> >
> > ----------
> >
> > On CIFAR10, we observed that MMA training is fairly stable to $d_\max$, and achieves good robustness-accuracy trade-offs.
> > With carefully chosen $\epsilon$ value, PGD-1.0/PGDLS-1.0 models can also achieve similar robustness-accuracy tradeoffs as compared to MMA training. However, when we perform PGD training with a larger $\epsilon$, the clean accuracy drops significantly.

---

> ### Author Response · Authors · 2019-11-11
> **Response to AnonReviewer3, part 2**
>
> >> 8. The proposed PGDLS is interesting, and similar to "dynamic training" in [3].
>
> Thank you for finding PGDLS interesting. We will add a discussion about [3] and PGDLS.
> Also, we are confused about that this item is a "con" for our paper. Note that PGDLS is only a stronger baseline we proposed in the paper. MMA is the main contribution of this paper. Could you please clarify the question if you do think there's a con here?
>
>
> >> 9. The gradient-free SPSA helps confirm the improvements of MMA under large perturbations are not a side effect of gradient masking.
>
> Thank you for the comment. Again, we are confused about that this item is a "con" for our paper. Could you please clarify the question if you do think there's a con here?

---

> ### Author Response · Authors · 2019-11-11
> **Response to AnonReviewer3, part 3**
>
> >> 5. Clarity of experimental settings. What CIFAR10-$\ell_{\infty}$ means? How the test attacks were generated? How the $m$ models were trained?
>
> Sorry about the confusion. Due to space limit, some important descriptions of the experiment settings are pushed to the appendix. We will modify the paper to make things clear in the main body.
>
> CIFAR10-$\ell_{\infty}$ means that the model is trained on the CIFAR10 dataset with $\ell_\infty$ attacks, and also tested with $\ell_\infty$ attacks on CIFAR10, as stated in Appendix C: "Here all the models are trained and tested under the same type of norm constraints, namely if trained on $\ell_\infty$, then tested on $\ell_\infty$; if trained on $\ell_2$, then tested on $\ell_2$."
>
> We take CIFAR10-$\ell_\infty$ as an example to explain the test settings. The complete list of models trained can be found in Table 3 (Appendix F), which contains 32 models ($m=32$). These include models trained with PGD/PGDLS with different $\epsilon$ (and their ensembles), models trained with MMA/OMMA with different $d_\max$, a standardly trained model, and the downloaded (Madry et al. 2018) PGD trained model.
> Assume that we want to evaluate the robustness of MMA-12 at perturbation magnitude 8/255. For each test example, we perform 10 PGD attacks with different random initializations (random starts) at this magnitude, and use the strongest attack among them to evaluate the robustness of the model, which is the typical "standard setting".
> At the same time, we also did this to all the 32 models that we trained, when we evaluate their robustness at perturbation magnitude 8/255. Therefore, we have $m \cdot N = 32\times 10 = 320$ attacks in total under 8/255, for each test example. For MMA-12, 10 of them are whitebox PGD attack with random starts, the other 310 attacks are transfer attacks, as they are generated from attacking other models on the same test example under the same perturbation magnitude.
> Therefore, for each model and each test example, we have 320 attacks, and if any one of them succeeds, we consider the model is not robust.
> Because of additional transfer attacks, our test setting is stronger than the standard setting. We also applied this testing protocol on the downloaded (Madry et al. 2018) model (bottom row of Table 3). Under CIFAR10-$\ell_\infty$ with perturbation magnitude 8/255, our test setting gives 44.68% robust accuracy, which is lower than 45.8% (originally reported in their paper).
> All the accuracies calcualted on the entire 10K test images.
>
> Reference:
> Madry et al. "Towards deep learning models resistant to adversarial attacks." ICLR 2018
>
>
> >> 5. How are the $d_\max$ determined? and what are their relationship to standard $\epsilon$?
>
> In MMA training, we try to maximize each example's margin until the margin reaches $d_\max$. In standard adversarial training, each example is trained to be robust at $\epsilon$.
> Therefore in MMA training, we usually set $d_\max$ to be larger than $\epsilon$ in standard adversarial training.
> It is difficult to know the "correct" value of $d_\max$, therefore we tested different $d_\max$ values. But different from PGD, MMA training is insensitive to this hyperparameter. A large $d_\max$ only slightly affect clean accuracy and robustness to small perturbations.
> In contrast, when $\epsilon$ is large in standard adversarial training, many examples that are not able to be $\epsilon$-robust are simply "gave up" (Figure 2, 4). Therefore clean accuracy and robust accuracies at small perturbations is largely impacted (Table 1).
>
> We will improve the clarity of the paper accordingly.

---

> ### Author Response · Authors · 2019-11-11
> **Response to AnonReviewer3, part 4**
>
> >> 2. which norm is $d_\theta$ defined on? concerns on $\delta^{*} = \arg\min_{\delta:L_\theta^{LM}(x+\delta, y) \geq 0} \|\delta\|$ in Theorem 2.1
>
> $d_\theta$ is defined on $\ell_p$ norm with arbitrary $p>0$, which includes both $\ell_\infty$ and $\ell_2$ norm. Therefore, we just used the general notation of norm $\|\cdot\|$.
>
> In Theorem 2.1, $\delta^{*}$ is not a norm, it is the $\delta$ that minimizes the norm $\|\delta\|$ under the constraint that $L_\theta^{LM}(x+\delta, y) \geq 0$.
> We don't think there is a mistake. Please let us know if the confusion remains.
>
>
> >> 3. The minimum margin \delta^{*} is a bit confusing, is it used in maximization or just in the outer minimization?
>
> In our terminology, $\delta^{*}$ is not the "minimum margin". It's norm $\|\delta^{*}\|=d_\theta(x, y)$ is the margin. $\delta^{*}$ is the shortest successful perturbation, which is the minimizer in Equation (2). $\delta^{*}$ is then used in the outer optimization in Equation (7).
>
>
> >> 3. inconsistency of $L(\theta, \delta)$ and $L(\delta, \theta)$
>
> Thanks for pointing this out. We will unify the notation to be $L(\delta, \theta)$.
>
>
> >> 4. Why do we need the "gradients of margins to model parameters" analysis from Proposition 2.1 to remark 2.2? Why don't go directly from Theorem 2.1 to Proposition 2.4?
>
> Theorem 2.1 is about margin maximization based on the LM loss, while Equation (7) is about maximizing a lower bound of the margin using the CE loss, we need some technical developments in between to make this transition.
> 1) Theorem 2.1 is a summarization of Proposition 2.1 and Proposition 2.2. You are right about the rest of section 2.1 after Theorem 2.1.
> It can be removed and the rest of the paper would not be affected. However, we believe this theoretical result is interesting and significant, since it rigorously shows how to **directly** maximize margin. This is arguably the most **direct** way of improving adversarial robustness under $\ell_p$ adversarial perturbation, but it was not discussed in literature to the best of our knowledge.
> Although minimizing the loss over $\delta^{*}$ seems an intuitive step from the perspective of adversarial training, we believe it is not obvious that the margin's gradient wrt $\theta$ is a scaled version of the loss' gradient wrt $\theta$ at $\delta^{*}$, as shown by Proposition 2.1.
> 2) Section 2.2, including contents before Proposition 2.4, is necessary for explaining the transition from the LM loss, which defines the margin, to the SLM/CE loss, which can be used to maximize margin's lower bound.

---

### Official Review · AnonReviewer1 · 2019-10-22
**Official Blind Review #1**

**Rating:** 6

**Review:**

This paper proposes a method, Max-Margin Adversarial (MMA) training, for robust learning against adversarial attacks. In the MMA, the margin in the input space is directly maximized. In order to alleviate an instability of the learning, a softmax variant of the max-margin is introduced. Moreover, the margin-maximization and the minimization of the worst-case loss are studied. Some numerical experiments show that the proposed MMA training is efficient against several adversarial attacks.

* review:
Overall, this paper is clearly written, and the readability is high. Though the idea in this paper is rather simple and straightforward, some theoretical supports are presented. A minor drawback is the length of the paper. The authors could shorten the paper within eight pages that is the standard length of ICLR paper.

- In proposition 2.4: the loss L^{CE} should be clearly defined.
- In equation (8), how is the weight of L^CE and L^MMA determined?

**Experience Assessment:**

I have read many papers in this area.

**Review Assessment: Checking Correctness Of Derivations And Theory:**

I did not assess the derivations or theory.

**Review Assessment: Checking Correctness Of Experiments:**

I did not assess the experiments.

**Review Assessment: Thoroughness In Paper Reading:**

I made a quick assessment of this paper.

---

> ### Author Response · Authors · 2019-11-11
> **Response to AnonReviewer1**
>
> Thank you for your comments. We are glad that you find the paper is clearly written and also value our theoretical results.
>
> >> A minor drawback is the length of the paper...
>
> We will try our best to further shorten the main body of the paper.
>
> >> In proposition 2.4: the loss $L^{CE}$ should be clearly defined.
>
> $L^{CE}_\theta = \log\sum_j \exp(f_\theta^j(x)) - f_\theta^y(x)$, and we will make it clear in the paper.
>
> >> In equation (8), how is the weight of $L^{CE}$ and $L^{MMA}$ determined?
>
> We tested 3 pairs of weights, (1/3, 2/3), (1/2, 1/2) and (2/3, 1/3), in our initial CIFAR10 Linf experiments. We observed that (1/3, 2/3), namely (1/3 for $L^{CE}$ and 2/3 for $L^{MMA}$) gives better performance.
> We then fixed it and use the same value for all the other experiments in the paper, including the MNIST experiments and L2 attack experiment s. We will make it clear in the appendix.

---

### Official Review · AnonReviewer2 · 2019-10-24
**Official Blind Review #2**

**Rating:** 6

**Review:**

Summary:
The paper propose to use maximal margin optimization for correctly classified examples while keeping the optimization on misclassified examples unchanged. Specifically, for correctly classified examples, MMA adopts cross-entropy loss on adversarial examples, which are generated with example-dependent perturbation limit. For misclassified examples, MMA directly applies cross-entropy loss on natural examples.

Problems:
1. For the performance measurement, why use the AvgRobAcc? does it make any sense to combine black-box results and white-box results?
2. For the epsilon, since it is different from the standard adversarial settings, how to guarantee the fair comparison? For example, how to evaluate the performance of  MMA-12 to PGD-8 under the same test attack PGD-8?
3. For the baseline, the authors lack some necessary baselines, like the following [1] and [2]
[1] Theoretically Principled Trade-off between Robustness and Accuracy. ICML 2019
[2] On the Convergence and Robustness of Adversarial Training. ICML2019

**Experience Assessment:**

I have published one or two papers in this area.

**Review Assessment: Checking Correctness Of Derivations And Theory:**

I assessed the sensibility of the derivations and theory.

**Review Assessment: Checking Correctness Of Experiments:**

I assessed the sensibility of the experiments.

**Review Assessment: Thoroughness In Paper Reading:**

I read the paper at least twice and used my best judgement in assessing the paper.

---

> ### Author Response · Authors · 2019-11-11
> **Response to AnonReviewer2**
>
> Thank you for your efforts in reviewing our paper and the questions. Please let us know if we do not directly address your concern in this response. We are happy to hear further feedbacks from you.
>
> Besides the contributions that you've summarized, we would also like to point out that
> 1) Our MMA training algorithm is not just a heuristic algorithm. The seemingly intuitive formulation of "minimizing cross-entropy loss on shortest successful perturbation" is backed up by our theories on **direct** margin maximization, and non-trivial construction of the margin's lower bound (Section 2).
> 2) Section 3 analyzes the how does the fixed $\epsilon$ in standard adversarial training influence training, from a margin maximization perspective. Our theoretical predictions are supported by results in Sections 4.
>
>
> >> 1. why use the AvgRobAcc?
>
> We believe AvgRobAcc, the average robust accuracy, is a more comprehensive measure than the robust accuracy under a fixed (and arbitrary) perturbation magnitude.
> In practice, it is difficult to argue to what attack magnitude, the robust accuracy is more important. When there is a tradeoff, it is difficult to decide if a model with higher robust accuracy to $8/255$ attacks but lower accuracy to $16/255$ attacks is more robust or less robust. Another example would be the tradeoff between robustness and clean accuracy. It seems more rasonable to measure the "area under the curve", which is approximated by AvgRobAcc.
>
> >> 1. does it make any sense to combine black-box results and white-box results?
>
> Our intention is to have the strongest attack on each model to approximate the "true" robustness of the model. Therefore we report robust accuracy against the strongest attack among both white-box and black-box attacks.
>
>
> >> 2. Fairness of evaluation
>
> Please see response to all reviewers.
>
> >> 3. For the baseline, the authors lack some necessary baselines, like the following [1] and [2]
>
> We are working on evaluating [1] and [2] under our test settings. We will report the results when ready.
>
> We would also like to make a comment that, as concurrent work, our idea on directly maximizing input space margin is orthogonal to [1]'s idea on optimizing a regularized surrogate loss, and [2]'s idea on dynamically adjusting the convergence of inner maximization.

---

> > ### Author Response · Authors · 2019-11-15
> > **Response to AnonReviewer2 on comparison to other baselines**
> >
> > Regarding [2], there is no publicly available pretrained models and we are not able to finish the training and evaluation before the rebuttal deadline. We will add the comparison to the paper later when ready. Nevertheless, [2]'s reported robustness is lower than those in TRADES [1], therefore we believe missing comparison to [2] will not affect the evaluation of our model significantly.
> >
> > ----------
> > Comparing to TRADES:
> > We evaluate the downloaded pretrained TRADES [1] models for both MNIST-$\ell_\infty$ and CIFAR10-$\ell_\infty$ cases. For each model, we show the clean accuracy and robust accuracies at different $\epsilon$'s.
> >
> > **MNIST models trained with $\ell_\infty$ attacks**
> >
> > MMA-0.45   acc: 98.9%, 0.1: 97.9%, 0.2: 96.0%, 0.3: 92.6%, 0.4: 85.2%
> >
> > TRADES     acc: 99.5%, 0.1: 98.8%, 0.2: 97.3%, 0.3: 94.1%, 0.4: 0.1%
> >
> > Overall TRADES outperforms MMA except that TRADES fails completely at large perturbation length of 0.4; This may be because TRADES is trained with attacking length $\epsilon = 0.3$. We are not able to train and evaluate a TRADES model with $\epsilon=0.4$ before the rebuttal deadline.
> >
> > **CIFAR10 models trained with $\ell_\infty$ attacks**
> >
> > MMA-32   acc: 84.4%, 4: 64.8%, 8: 47.2%, 12: 31.5%, 16: 18.9%, 20: 10.2%, 24: 4.8%, 28: 2.0%, 32: 0.8%
> >
> > TRADES    acc: 84.9%, 4: 71.0%, 8: 52.9%, 12: 33.0%, 16: 18.2%, 20: 8.3%, 24: 3.6%, 28: 1.4%, 32: 0.7%
> >
> > Here we compare MMA-32 to TRADES because their clean accuracies are similar. Similar to the results in MNIST, TRADES outperforms MMA on the attacks of lengths that are less than $\epsilon = 12/255$, but it sacrifices the robustness under larger attacks.
> >
> > We would also like to reiterate that the focus of this paper is not to achieve the best robustness toward some fixed attack magnitude.
> > Our contribution is a direct margin maximizing perspective for adversarial robustness, and it is connection to adversarial training.
> > Based on our analysis, we propose MMA that can achive robustness across different attacking lengths based on their intrinsic robustness.
> > Our idea and the idea in TRADES of optimizing a calibration loss are progress in different directions and could potentially be combined.

---

> > > ### Comment · AnonReviewer2 · 2019-11-15
> > > **Thanks for your response**
> > >
> > > Thanks for your clarification and response. I am happy with your answers, and have upgraded the rating to 6 (Weak Accept).

---

### Public Comment · ~Anthony_Wittmer1 · 2019-09-30
**Great work.**

It is a great work.

Since PGD adversarial training only train with the adversarial exmamples, when the epsilon becomes larger, it will be hard to converge. Thus, the models PGD-24 and PGD-32 are hard to train.

In Table 1, the results of PGD-24 show bad performance on both clean data and avdersarial data, which reveals PGD-24 may need a stronger neural network.

The training of PGDLS-24 can converge, as the its learning process is from the easy way to the difficult way, by linearly increasing the epsilon.

Have the authors tried another stronger baseline, TRADES[1], to compare with, such TRADES-24 and TRADES-32. For TRADES, it trains with both clean data and adversarial data via surrogate-loss minimization. So I think it will converge and show less sensitivity to hyperparameter setting when the epsilon becomes large.

[1] Theoretically Principled Trade-off between Robustness and Accuracy. ICML 2019

---

> ### Author Response · Authors · 2019-10-04
> **thank you and wrt TRADES**
>
> Thank you for your kind comment.
>
> We haven't tried training TRADES with a larger epsilon yet. Training with clean data is essentially training with epsilon=0. According to the theory in our paper, for already correctly classified data, training on clean data is also "adversarial training with an epsilon smaller than the margin", so it maximizes a lower bound (although loose) of the margin. Combining this with the training on "adversarial data via surrogate-loss minimization", it could be possible that TRADES is less sensitive to hyper-parameter settings wrt large epsilon values, and TRADES-24/32 converges.
>
> However, we note that even if TRADES-24/32 converges, all the data points are still using a single epsilon, and thus 1) the choice of epsilon is arbitrary; 2) fixing epsilon does not consider that different data points might have different intrinsic robustness. Since our idea is orthogonal to the idea of TRADES, they could potentially be combined for further improvements.

---

### Author Response · Authors · 2019-11-11
**Response to All Reviewers**

We thank all reviewers for their efforts in review and thoughtful comments.

Here we address the common concern from R2 and R3 about fairness in evaluation.

>> R2: 2. For the epsilon, since it is different from the standard adversarial settings, how to guarantee the fair comparison? For example, how to evaluate the performance of  MMA-12 to PGD-8 under the same test attack PGD-8?
>> R3: 6. Fairness of the comparison. Since MMA changes $\epsilon$, how to fairly compare the robustness to standard epsilon bounded adversarial training is not discussed. Is it fair to compare MMA-3.0 vs PGD-2.5, since they have different epsilon?

We believe the comparison is fair for 3 reasons: 1) the test settings are strong enough and are the same for all models, regardless of how they are trained; 2) Due to the different meanings of $d_\max$ and $\epsilon$, it is not clear what value of $d_\max$ is a fair comparison to $\epsilon = 8/255$. Instead, in the paper we compared a group of MMA trained models to a group of PGD trained models with different $d_\max$'s and different $\epsilon$'s (or the best from MMA to the best from PGD); 3) because PGD trained models are sometimes tested on the same PGD attack used for training, the evaluation is at least not in favour of MMA training.

To elaborate on the first two points:
1) Regardless of how a model is trained, we care about whether it is robust at test time. We believe it is a fair comparison for different models, as long as they are evaluated under the **same** test setting, and the testing attacks are strong enough for each model.
Note that although our training algorithm is different from the standard adversarial training, we do use the same **standard adversarial test settings**, i.e. evaluating robust accuracies under repeated PGD attacks (both whitebox and transfer) at different perturbation magnitudes.
Specifically, although trained with different algorithms, when we compare MMA-12 and PGD-8 models wrt their robustness at 8/255 perturbation magnitude for the CIFAR10-$\ell_\infty$ case, we believe that the same testing protocol with repeated PGD-8 attacks is strong enough on both MMA-12 and PGD-8, therefore it is a fair comparison.

2) Since MMA training and PGD training have different types of hyperparameters, one-to-one comparison between MMA trained and PGD trained models might not be fair, e.g. MMA-12 vs PGD-8. However, in our evaluation, we trained a group of models for both MMA training and PGD training, covering reasonable values of $d_\max$ and $\epsilon$. Our comparison is between the MMA group and PGD group.
(To R3:  Your comment might be related to Figure 4. Our intention was to verify the theory that MMA will be able to uniformly increase the margins of different data points while PGD cannot. PGD-2.5 vs MMA-3.0 is an arbitrary choice for comparison. In Figure 4, we are interested in the qualitative analysis of the pattern, rather than a concrete quantitive metric.)

---

### Decision · Program_Chairs · 2019-12-19

**Decision:**

Accept (Poster)

**Comment:**

This work presents a new loss function that combines the usual cross-entropy term with a margin maximization term applied to the correctly classified examples. There have been a lot of recent ideas on how to incorporate margin into the training process for deep learning. The paper differs from those in the way that it computes margin. The paper shows that training with the proposed max margin loss results in robustness against some adversarial attacks.
There were initially some concerns about baseline comparisons; one of the reviewers requesting comparison against TRADES, and the other making comments on CW-L2. In response, authors ran additional experiments and listed those in their rebuttal and in the revised draft. This led some reviewers to raise their initial scores. At the end, majority of reviewers recommended accept. Alongside with them, I find extensions of classic large margin ideas to deep learning settings (when margin is not necessarily defined at the output layer) an important research direction for constructing deep models that are robust and can generalize.